# Tumor detection by analysis of both symmetric- and hemi-methylation of plasma cell-free DNA

Xu Hua[1,2,3,4,10], Hui Zhou[1,2,3,4,10], Hui-Chen Wu[2,5], Julia Furnari[2,6], Corina P. Kotidis [2,7], Raul Rabadan [8], Jeanine M. Genkinger [2,9], Jeffrey N. Bruce [2,6], Peter Canoll [2,7], Regina M. Santella[2,5] & Zhiguo Zhang [1,2,3,4] ✉

Aberrant DNA methylation patterns have been used for cancer detection. However, DNA hemi-methylation, present at about 10% CpG dinucleotides, has been less well studied. Here we show that a majority of differentially hemi-methylated regions (DHMRs) in liver tumor DNA or plasma cells free (cf) DNA do not overlap with differentially methylated regions (DMRs) of the same samples, indicating that DHMRs could serve as independent biomarkers. Furthermore, we analyzed the cfDNA methylomes of 215 samples from individuals with liver or brain cancer and individuals without cancer (controls), and trained machine learning models using DMRs, DHMRs or both. The models incorporated with both DMRs and DHMRs show a superior performance compared to models trained with DMRs or DHMRs, with AUROC being 0.978, 0.990, and 0.983 in distinguishing control, liver and brain cancer, respectively, in a validation cohort. This study supports the potential of utilizing both DMRs and DHMRs for multi-cancer detection.

Cancer is a major public health threat worldwide. While the cancer death rate has fallen continuously since the peak in 1991 in the United States, it is estimated that over 600,000 people died from cancer in 2021 in the United State alone[1]. World-wide, almost 10 million people died from cancer in 2020, and the death rate increased in some low and middle income countries in recent years[2]. Therefore, there remains an urgent and unmet need to combat cancer. It has been shown that early tumor detection has potential to improve prognosis for cancer patients[3]. For instance, the five-year survival rate for hepatocellular carcinoma (HCC) when diagnosed at early and localized stage is 34%, but drops to 3% when diagnosed at late stage with distant disease[1]. Early cancer detection also contributed to the reduced cancer death rate in the United States in the last couple of decades. Therefore, it is critically important to develop assays for early cancer detection.

It has been proposed that liquid biopsy offers several advantages for cancer early detection[4–8]. For instance, liquid biopsy samples can be obtained non-invasively and in principle can overcome the challenges arising from tumor heterogeneity that confounds tissue biopsy procedures. Indeed, several methods have been developed to use

[1]Institute for Cancer Genetics, Columbia University Irving Medical Center, New York, NY 10032, USA. [2]Herbert Irving Comprehensive Cancer Center, Columbia University Irving Medical Center, New York, NY 10032, USA. [3]Department of Pediatrics, Columbia University Irving Medical Center, New York, NY 10032, USA. [4]Department of Genetics and Development, Columbia University Irving Medical Center, New York, NY 10032, USA. [5]Department of Environmental Health Sciences, Mailman School of Public Health, Columbia University, New York, NY 10032, USA. [6]Department of Neurological Surgery, Columbia University Irving Medical Center, New York, NY 10032, USA. [7]Department of Pathology and Cell Biology, Columbia University Irving Medical Center, New York, NY 10032, USA. [8]Program for Mathematical Genomics and Department of Systems Biology, Columbia University Irving Medical Center, New York, NY, USA. [9]Department of Epidemiology, Columbia University Mailman School of Public Health, New York, NY, USA. [10]These authors contributed equally: Xu Hua, Hui Zhou. ✉e-mail: zz2401@cumc.columbia.edu

plasma cell free (cf) DNA for tumor detection. Plasma cfDNA are a mixture of extracellular DNA fragments released from apoptotic and/ or necrotic cells or released via active secretion[6,7]. While the majority of the plasma cfDNA comes from normal cells such as lymphocytes, cancer cells also release DNA fragments into circulation[9]. Analysis of cancer related mutations in plasma cfDNA has been reported for early cancer detection. Due to limited mutations in cancer cells and the evolving nature of these mutations, a significant amount of test material (7.5 – 10 ml plasma) as well as sequence depth are required for specific mutation detection[10,11]. In addition to genetic mutations, other cfDNA features including fragmentomics[12,13], and epigenetic features including nucleosome patterns[14–16] and DNA methylation[17–22] have also been analyzed for detection of a variety of cancer types. Among all these features analyzed so far, it was reported that models trained with DNA methylation performed better in tumor detection than models trained using other features including single nucleotide variants or cfDNA pan features including fragment length[23], supporting the idea that analysis of DNA methylomes of plasma cell free DNA likely represents an outstanding approach for tumor detection.

The vast majority of DNA methylation in mammalian cells occurs at CpG dinucleotides in a symmetric manner: the cytosines (C) in a CpG dinucleotide on both Watson strand and its complementary Crick strand are methylated[24,25]. During DNA replication, hemi-methylated CpG dinucleotides, consisting of methylated CpGs on the parental strand and non-methylated CpGs on the complementary nascent strand, are rapidly converted into symmetric and fully methylated CpGs to maintain DNA methylation patterns[25]. Early studies indicate that failure of symmetrical methylation and inter-mediates of active demethylation in cancer genesis contributes to the generation of DNA hemi-methylation (HM)[26]. Interestingly, it has been observed that about 10% of CpG dinucleotides in human embryonic stem cells are hemi-methylated, and these hemi-methylated regions (HMRs) could be maintained during multiple cell divisions[27,28]. Moreover, it has been shown recently that while HM at the motif strand of CTCF, a critical regulator of genome organization, inhibits the binding of CTCF, HM on the opposite strand stimulates CTCF binding[29]. Therefore, HM is an epigenetic mark, and likely plays an important role to regulate genome organization and gene transcription. However, while HMRs have been analyzed in various cell lines based on bisulfite-sequencing (BS-Seq) or MeDIP-Seq[30–35], few studies, if any, have explored these hemi-methylated regions alone or in combination with symmetrically methylated CpGs for tumor detection and for tumorigenesis.

Currently, various methods have been used to analyze cfDNA methylomes for cancer detection. For instance, the CCGA (Circulating Cell-Free Genome Atlas) employed targeted bisulfite sequencing to analyze methylated regions of cfDNA in more than 50 cancer types[21,22]. Because bisulfite treatment of DNA results in a marked loss of DNA, this method, on average, requires up to 8 – 10 ml plasma and over 100 million sequence reads per sample for tumor detection. Shen et al.[20] and Nassiri et al.[19] developed a cell-free methylated DNA immuno-precipitation and high-throughput sequencing (cfMeDIP–seq) method based on double strand DNA ligation. This method requires cfDNA purified from 0.5 – 3.5 ml plasma samples. Because a fraction of cfDNA molecules are single-stranded (ss) DNA fragments and damaged double-stranded DNA fragments, these DNA molecules are in principle not used for DNA methylome analysis based on the traditional double stranded DNA based library preparation method. It has been shown that single-stranded DNA (ssDNA) library preparation method can include all cfDNA molecules (ssDNA, dsDNA and damaged DNA) for sequencing analysis, which leads to increased sensitivity compared to methods using traditional library preparation methods[36,37]. Therefore, in principle, utilization of ssDNA library preparation method for cfDNA methylome analysis will likely increase the sensitivity compared to methods relying on double stranded DNA ligation. Importantly,

none of these studies utilized DNA hemi-methylation for tumor detection, likely due to the fact that it was not known whether differentially hemi-methylated regions (DHMRs) are independent biomarkers from differentially methylated regions (DMRs).

We developed two methylated DNA immunoprecipitation and strand-specific (ss) sequencing methods (MeDIP-Seq) for genomic DNA (ssg-MeDIP-Seq) and plasma cell free (cf) DNA (sscf-MeDIP-Seq) for analysis of methylomes, respectively. The sscf-MeDIP-Seq method can analyze methylomes of cfDNA molecules including ssDNA, dsDNA and damaged DNA. Therefore, we produced reliable sscf-MeDIP-Seq datasets with cfDNAs isolated from 300 – 500 µl of plasma samples. Importantly, both methods can analyze both symmetrically methylated and hemi-methylated regions. Through in-depth analysis of DMRs and DHMRs of liver tumor DNA and cfDNA samples, we found that the vast majority of tumor DNA as well as cfDNA DHMRs do not overlap DMRs for the same samples, suggesting that DHMRs can serve as biomarkers independent of DMRs. Indeed, we found that machine learning models using both DMRs and DHMRs as input features outperform models trained with DMRs or DHMRs alone for tumor detection based on analysis of 271 plasms cfDNA samples. Together, our studies reveal that the utilization of both DMRs and DHMRs identified by the sscf-MeDIP-seq procedures as biomarkers will likely improve the accuracy for multi-cancer detection.

## Results

### Develop genomic methylated DNA immunoprecipitation with a strand-specific sequencing method (ssg-MeDIP-Seq)

MeDIP-seq has been used to analyze DNA methylation (5-mC), and almost all published MeDIP-Seq procedures rely on sonication of genomic DNA into small fragments followed by immunoprecipitation with antibodies against methylated DNA[35]. As Tn5 transposase has been used for genomic DNA fragmentation for the generation of libraries for next generation sequencing, we tested whether Tn5 can be used for fragmentation of genomic DNA before immunoprecipitation (Fig. 1a). Briefly, 100 ng of genomic DNA isolated from tissues were incubated with pA-Tn5 transposase, which fragments and inserts an adaptor into dsDNA in a sequence independent manner. As pA-Tn5 transposase covalently ligates the adaptor to the 5' end of target DNA, we then ligated a different adaptor at the 3' end through the oligo-replacement step. In this way, we could analyze DNA methylation patterns in a strand-specific manner, which in turn allows us to detect both symmetric DNA methylation (SM) as well as hemi-methylation (HM). We termed this method as ssg-MeDIP-Seq. Following the adaptor ligation, DNA fragments were denatured into single-stranded DNA (ssDNA) and methylated DNAs were immunoprecipitated using antibodies against 5-mC. The enriched methylated ssDNAs were amplified by PCR for library preparation and subsequent sequencing (Fig. 1a). Using this method, we first analyzed DNA methylation of 16 tissue samples, eight isolated from liver tumors and eight from their corresponding adjacent non-tumor (Adj-NT) tissues. The ssg-MeDIP-seq signals of the 8 tumor samples were depleted at the promoters of genes with CpG island (CGI) compared to those without CGI (Supplementary Fig. 1a), a pattern consistent with DNA methylation detected using other methods. A similar pattern was also detected for plasma cfDNA methylation analyzed by sscf-MeDIP-seq described below (Supplemental Fig. 1b–d). Next, by comparing methylomes of eight liver tumors to their corresponding adjacent non-tumor tissues at 2,002,724 DNA methylation blocks, which cover 70% of CpG dinucleotides in the genome, with each block consisting of at least four CpGs[38], we identified 11,930 hypermethylated DMRs and 12,974 hypomethylated DMRs (Fig. 1b). For instance, a DMR specifically in tumors compared to Adj-NT samples was identified at the gene locus of *TBX2*, a gene known to be methylated in liver cancer[39] (Fig. 1c). To determine whether these DMRs identified in liver tumors showed

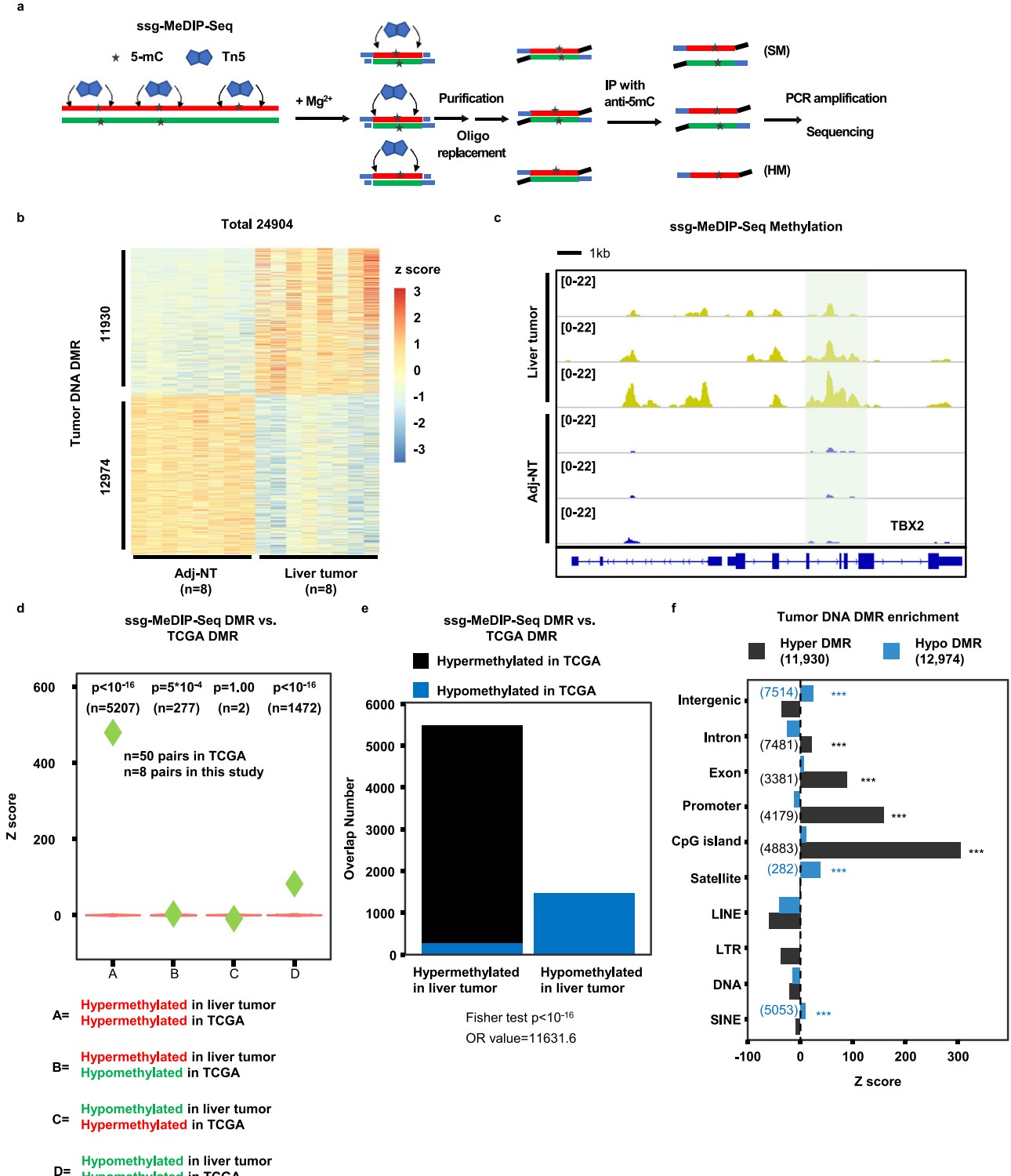

concordance with DMRs of liver tumors from an independent source, we analyzed the DNA methylation profiles of 50 liver cancer sample from TCGA, which were generated using 450 K CpGs methylation microarray but were not suitable for analysis of HM regions (HMR) (see below). Despite the dramatic technical differences between ssg-MeDIP-Seq and 450 K methylation arrays, we found that hypomethylated and hypermethylated DMRs identified in liver tumors using ssg-MeDIP-seq overlapped significantly with hypomethylated and hypermethylated DMRs identified using the TCGA liver cancer datasets, respectively (group "A" and "D" in Fig. 1d). In contrast, concordance

between hypermethylated DMRs identified using ssg-MeDIP-Seq and hypomethylated DMRs in the TCGA datasets and vice versa was not so significant (group "B" and "C" in Fig. 1d). Similar results were obtained to analyze the overlaps between liver tumor DMRs from this study and by TCGA using the Fisher test (Fig. 1e). Finally, we found that hypermethylated DMRs were enriched at exons, promoters and CGIs, whereas hypo-methylated DMRs were enriched at intergenic regions, satellites, and SINEs (Fig. 1f). Taken together, these results indicate that the ssg-MeDIP-seq procedure can be used for analyzing genomic DNA methylomes.

**Fig. 1 | A pA-Tn5-based MeDIP-Seq method for analyzing methylomes of genomic DNA in strand-specific manner. a** An outline of ssg-MeDIP-Seq procedures for the analysis of DNA methylation of genomic DNA in a strand-specific manner. SM symmetric methylation, HM hemi-methylation. **b** Heatmap of DMRs between 8 liver tumor and corresponding Adj-NT. Z score, shown in color, represents the log$_2$ (RPKM) value of ssg-MeDIP-Seq signals. **c** A snapshot of liver tumor DNA DMR at the *TBX2* gene locus of three liver cancer samples and their corresponding adjacent non-tumor (Adj-NT) tissue samples. The shading area highlighted DMR identified by 8 liver tumor samples compared to 8 Adi-NT controls, with three samples shown here. The other regions in the snapshot were not significant based on QSEA. Overlaps between liver cancer DMRs identified by ssg-MeDIP-Seq in this study and those from TCGA tumor samples by 450 K methylation arrays using violin plots (**d**) and bar plots (**e**). Violin plots represent the random distribution of overlaps from 100 permutations, and *P* values were computed by the random permutation distribution in a one-sided way. Green diamonds represent observed overlap between DMRs identified from TCGA liver tumors and DMRs identified by ssg-MeDIP-Seq. The statistical analysis for the bar plot was performed using the Fisher test in a two-sided way. **f** The sequence element enrichment of liver tumor DNA DMRs. The DMRs were first overlapped with each annotated locus and compared with the overlapped number in random distribution for the calculation of the Z score. The *p* value was computed by the random distribution in a one-sided way and no multiple comparison correction was performed. The significantly enriched sequence elements were labelled with asterisks shown with color of black (hyper-methylated DMR) and blue (hypo-methylated DMR), with the number of DMRs in each category shown in the parenthesis. (*$p < 0.05$; **$p < 0.01$; and ***$p < 0.001$). LINE, Long Interspersed Nuclear Element retrotransposons; LTR long-terminal repeat, SINE short interspersed nuclear element, DNA DNA transposon.

## Liver tumor DNA DHMRs and DMRs are likely independent biomarkers

Recently, it has been shown about 10% of CpG dinucleotides are hemi-methylated (Fig. 2a), and are heritable[27,28]. However, to our knowledge, no studies have been performed to compare DHMRs to DMRs for the same samples systematically. Because ssg-MeDIP-Seq method could detect DNA methylation at Watson and Crick strands separately, we therefore analyzed the hemi-methylated regions (HMRs) at 2,002,724 blocks[38] in 8 liver tumor samples and their matched Adj-NT using the formula shown in Fig. 2a. To minimize the contribution of the experimental procedures and sequence depth to the identification of false positive HMRs, we first prepared libraries of two input samples, one liver tumor and one Adj-NT, by following the same procedure of ssg-MeDIP-Seq except that these two DNA samples were not subjected to methylated DNA immunoprecipitation. In principle, these input samples should not exhibit HM at the ~2 M methylation blocks. Indeed, majority of ~2 M blocks did not show strand bias signals (Supplemental Fig. 2a, b). In contrast, a marked number of blocks showed strand bias signals/HM signals based on the cutoff of Watson-Crick)/(Watson+Crick)>0.3 in 16 ssg-MeDIP-seq samples (Supplemental Fig. 2c, d). Because HM signals at each block were calculated using the formula, (Watson-Crick)/(Watson+Crick), sequence depth may affect HMR identification. Therefore, we tested different RPM at each block as additional cutoffs. We found that the number of blocks showing strand bias for the two input samples was reduced dramatically using sequence read RPM > 1 at each block as the cutoff compared to that RPM > 0.5 (Supplemental Fig. 1b). A further increase of RPM to 1.5 or 2 as the cutoff did not reduce the number of blocks showing bias markedly (Supplemental Fig. 1b). Similar results were found when we analyzed 8 input samples of plasma cell free DNA (Supplemental Fig. 1e, f, see below). Therefore, we used the cutoff ((Watson-Crick)/(Watson+Crick)>0.3, RPM > 1, and *p* < 0.01) to identify HMRs of 8 liver tumor DNA and their corresponding Adj-NT, and identified 192,106 and 228,575 HMRs in 8 liver tumor and their Adj-NT groups, respectively. The number of HMRs identified in both group of samples was roughly ~10% of ~2 M methylation blocks used for analysis. Furthermore, the HMRs of both liver tumor and Adj-NT were enriched the most at genomic regions of SINEs, CpG islands, promoters and exons, and with a slight enrichment at satellites and introns (Fig. 2c). Finally, we identified 6864 DHMRs in liver tumor DNA samples compared to their corresponding Adj-NT. These DHMRs included 2330 regions with increased HM and 4534 regions with reduced HM at either Watson or Crick strands compared to the controls (Fig. 2d). Remarkably, the majority of liver tumor DHMRs (4474 out of 6562) did not overlap with DMRs (Fig. 2e). The DHMRs with increased HM in liver tumor samples were enriched at genomic regions of SINEs, CpG islands, promoters and exons, whereas DHMRs with reduced HM were enriched at SINEs and CpG islands, but not promoters (Fig. 2f). Interestingly, the closest genes within 20 kb to these liver tumor HMRs (Fig. 2g) and DHMRs with increased HM (Fig. 2h) were enriched in processes linking to

cellular metabolism. These results suggest that DHMRs likely represent independent biomarkers, consistent with the idea that DNA hemi-methylation is an epigenetic marker.

To understand why the majority of DHMRs did not overlap with DMRs, we analyzed the methylation density at either Watson or Crick strands at 24,904 DMRs and 6864 DHMRs of the 8 liver tumor samples compared to their Adj-NTs. We found that the methylation density at only one strand (either Watson or Crick strands) of 6864 DHMRs was increased or reduced markedly in tumor samples compared to Adj-NT controls (Supplemental Fig. 3a–d). In contrast, the methylation density of both Watson and Crick strands at DMRs were increased or reduced to a similar degree in liver cancer samples compared to the same Adj-NT controls (Supplemental Fig. 3e–h). These results suggest that DHMRs arise from changes in DNA methylation at one strand, whereas DMRs from changes in DNA methylation of both strands. Taken together, these results indicate that liver tumor DHMRs and DMRs are most likely independent biomarkers.

## Develop the sscf-MeDIP-Seq method for analyzing cfDNA methylation and hemi-methylation

There is a tremendous interest in analyzing plasma cfDNA methylomes for tumor detection[20]. Compared to the large size of genomic dsDNA, plasma cfDNAs are a mixture of dsDNA and ssDNA with major fragment sizes about 160 – 170 nucleotides. Furthermore, some of these DNA are nicked or damaged[36]. We took advantage of our extensive experience in preparing single-stranded DNA libraries for next-generation sequencing[40,41], which originated from methods for sequencing ancient DNA samples also consisting of dsDNA, ssDNA and damaged DNA[42], to develop procedures to analyze cfDNA methylomes. Briefly, after denaturing cfDNA into ssDNA, we ligated an adaptor to the 3' end of cfDNA using an ssDNA ligase followed by converting ssDNA into dsDNA by a DNA polymerase. After the ligation of the second adaptor, a small fraction of DNA (10%) was saved as the input sample, and the remaining DNA was denatured again and subjected to immunoprecipitation using antibodies against 5-mC. The immunoprecipitated DNA as well as the input DNA were then amplified by PCR for library preparation and sequencing (Fig. 3a). In this way, all DNA molecules including dsDNA, ssDNA, and damaged DNA will be utilized for methylome analysis (Fig. 3a). Importantly, this method can analyze both SM and HM. We termed the method as single-stranded (ss)cf-MeDIP-Seq.

Using this method, we first performed in-depth analysis of 20 sscf-MeDIP-Seq datasets generated from cfDNAs of 10 individuals with liver tumors and 10 controls with similar age and gender distributions to gain insight into the performance of sscf-MeDIP-Seq and the properties of cfDNA DMR and DHMRs. Similar to ssg-MeDIP-seq, sscf-MeDIP-seq signals were depleted at the promoter regions of genes with CGI compared to those without CGI for all the three sample groups (Supplemental Fig. 1b–d). Using the same 2,002,724 methylation blocks[38], we identified 2229 hyper-methylated and 5002 hypo-methylated

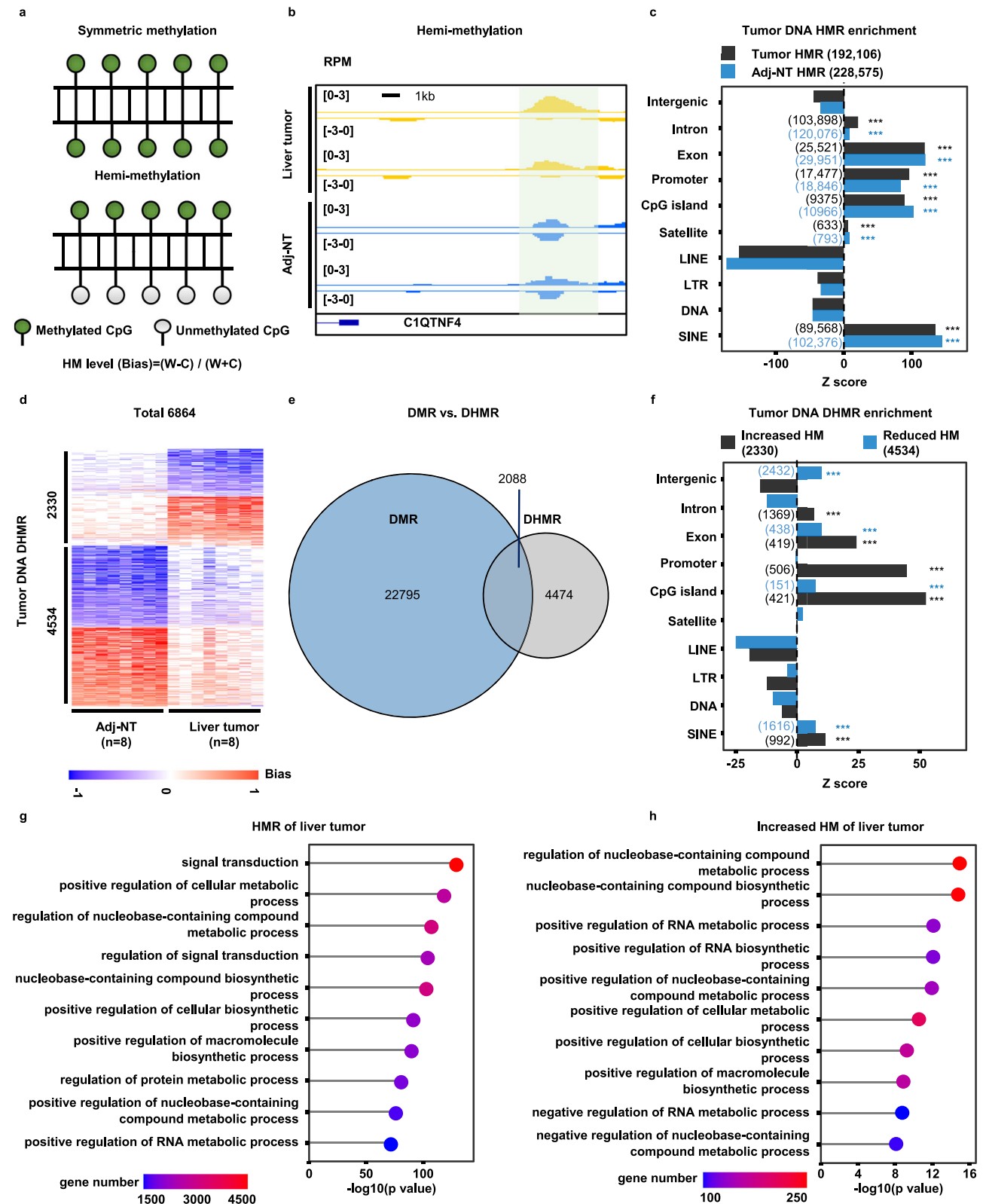

cfDNA DMRs for 10 liver cancer cfDNA samples compared to the 10 controls (Fig. 3b, c), with hyper-methylated cfDNA DMRs enriched at CGIs, promoters and exons and hypo-methylated ones at satellite DNA, intergenic regions, CGI and SINEs (Supplemental Fig. 4a). We then asked whether these liver cancer cfDNA DMRs overlapped with liver tumor DNA DMRs identified in Fig. 1. We found that both hyper-methylated and hypo-methylated cfDNA DMRs exhibited significant

overlap with liver tumor DNA hyper-methylated and hypo-methylated DMRs analyzed by ssg-MeDIP-Seq, respectively ("A" and "D" group in Fig. 3d). In contrast, the overlaps between hypo-methylated cfDNA DMRs and hyper-methylated liver tumor DNA DMRs or vice versa ("B" and "C" groups Fig. 3d) was much less significant. Similar analysis of overlaps between cfDNA DMRs and liver tumor DNA DMRs using the Fisher test showed the same conclusion (Fig. 3e). Together, these

**Fig. 2 | Analysis of DNA hemi-methylation of liver cancer samples by ssg-MeDIP-Seq. a** An illustration of symmetric methylation (SM) and hemi-methylation (HM). SM refers to DNA methylation at CpG dinucleotides of both Watson and Crick strands equally, where a HM region (HMR) refers to preferential methylation of CpG dinucleotides at one strand over the other strand. HM level (bias) was calculated using (W-C)/(W + C). W and C represent sequence reads at Watson and Crick strand at each block. **b** A snapshot of tumor DNA differentially hemi-methylation region (DHMR) at the *C1QTNF4* gene locus of two liver tumor samples compared to their corresponding Adj-NT, with the shaded area indicating a DHMR. Please note the RPM values, but not the calculated HM level/bias, were shown. **c** The sequence enrichment for liver tumor DNA HMRs. The HMRs were calculated using formula shown in (**a**) and identified using cutoff described in the text. The Z score was calculated by compared with the overlapped number in random distribution. The significantly enriched sequence elements were labelled with asterisks, with HMRs for liver tumor and Adj-NT DNA shown as black and blue, respectively. The number of HMRs in each element was shown in the parenthesis. **d** Heatmap of total 6864 DHMRs from 8 liver tumor samples compared to their corresponding Adj-NT. HM level is shown in color from −1 to 1, with 2330 liver tumor DNA DHMRs showing increased HM at either Watson or Crick strand, and 4,534 DHMRs showing reduced HM compared to controls. **e** Overlap of DMRs and DHMRs of eight liver tumor samples compared to their corresponding Adj-NT. **f** The enrichment for liver tumor DNA DHMRs of increased (black) and reduced (blue) HM compared to controls samples. **g** The GO function enrichment for genes closest to liver tumor DNA HMRs. **h** The GO function enrichment for genes closet to liver tumor DNA DHMRs with increased HM compared to Adj-NT control samples. The *p* values in c and f were computed by the random distribution in a one-sided way and no multiple comparison corrections were performed (*$p < 0.05$; **$p < 0.01$; and ***$p < 0.001$). For **g** and **h**, the GO enrichment was tested by cumulative hypergeometric *p* values in a one-sided way and performed by R package "gprofiler2". The multiple comparison correction was made by the "set counts and sizes" method.

results show that plasma cfDNA DMRs of patients with liver tumor identified by sscf-MeDIP-Seq method most likely reflect DNA methylation changes in liver cancer cells.

## The majority of plasma cfDNA DHMRs also do not overlap with cfDNA DMRs

To identify cfDNA DHMRs, we first analyzed 8 input samples that were prepared by following the same procedure as sscf-MeDIP-seq except that these 8 DNA samples were not subjected to methylated DNA immunoprecipitation. We found that like the two genomic DNA input samples (Supplemental Fig. 2a, b), the number of blocks that exhibited strand bias was markedly reduced using RPM > 1 at each block as the cutoff compared to RPM > 0.5 (Supplemental Fig. 2e, f). We therefore used the same cutoff for the analysis of the ssg-MeDIP-Seq datasets and analyzed cfDNA DHMRs of these 10 liver tumor samples compared to the 10 controls and identified 1179 and 988 DHMRs with increased and reduced HM at either Watson or Crick strand, respectively, compared to the 10 control samples (Fig. 3f). These cfDNA HMRs from both liver cancer and control samples were enriched at SINEs, satellites, promoters and exons (Supplemental Fig. 4b). In contrast, cfDNA DHMRs specific for liver tumor samples with increased HM were enriched at CpG islands, promoters and exons, whereas those with reduced HM were enriched at SINEs, exons and intergenic regions (Supplemental Fig. 4c). Finally, we asked whether liver tumor cfDNA DHMRs also showed a significant overlap with liver tumor DNA DHMRs when compared with the same control cfDNA samples. We observed that cfDNA DHMRs with increased and reduced HM showed significant overlap with tumor DNA DHMRs with increased and reduced HM, respectively (Supplementary Fig. 4d, e). These results indicate that cfDNA DHMRs likely also reflect tumor DNA DHMRs. Importantly, like liver tumor genomic DNA DMRs and DHMRs, the vast majority of plasma cfDNA DHMRs from liver cancer samples did not overlap with cfDNA DMRs for the same samples (Fig. 3g), indicating that cfDNA DHMRs could also be used as independent biomarkers for tumor detection.

## Identification of cancer types using machine learning models trained using DMRs, DHMRs and DMRs+DHMRs as inputs

It has been shown that cfDNA methylation could be used to identify tumor origins[18]. To determine whether sscf-MeDIP-Seq procedures could be used for tumor prediction, we analyzed cfDNA methylomes of three groups of plasma samples: patients with liver (73 samples) or brain (97 samples) cancer and controls (101 samples) (Table 1) and generated a total 271 sscf-MeDIP-Seq datasets. Of the 271 sscf-MeDIP-Seq datasets generated, 215 datasets including 58 liver cancer and 77 brain cancer samples, and 80 controls were randomly selected and used as the training cohort to train machine learning models of GLMnet, random forest or deep neural network (DNN) (Fig. 4a). All three machine learning models accurately predicted samples in the validation cohorts (56 samples consisting of 20 brain cancer, 15 liver cancer and 21 control samples), with GLMnet models showing the best performance (Fig. 4b–e and Supplemental Fig. 5a–f), highlighting the robustness of our prediction and sscf-MeDIP-Seq datasets. As general procedures for model training and sample validation are similar for all three models, we focused our discussion on GLMnet models below.

To reduce the influence of diversity of individual samples on model training, we randomly sampled 90% of the samples in the training cohort 10 times in a balanced way (control, brain and liver cancer), identified cfDNA DMRs and DHMRs specific for each sample group in a one-versus-other way, and selected the top DMRs and DHMRs based on the feature importance determined by the GLMnet models. In the beginning, we trained these models using different DMRs and DHMRs of each sample group with DMRs selected by *p* value and log fold change (LFC) of DNA methylation density and DHMRs selected by feature importance defined by the GLMnet models. We observed an increase in model performance when more stringent parameters were used for DMR and DHMR selection (Supplemental Fig. 5g–i). In the end, we selected the top 200 DMRs and 200 DHMRs from the three sample groups for each of the 10 rounds of training using either DMRs or DHMRs as inputs (Fig. 4a). We then combined DMR and DHMR models to train a calibration model for the final prediction of each sample in the training cohort. Briefly, to predict sample identity in the 56-sample validation cohort, we first predicted each sample using 10 models trained with DMRs or DHMRs as the inputs, and then combined the prediction results as the inputs of the calibration model to obtain final prediction probability of each sample. In general, we observed that models based on DMRs alone were slightly better predictors than models based on DHMR alone (Fig. 4b–d). Furthermore, when combined, DMR+DHMR-based models yielded a slightly more accurate prediction than models based on either DMRs or DHMRs alone (Fig. 4b–d), with AUROC of models using both DMR and DHMR as inputs for brain cancer, liver cancer and controls being 0.983 (95% confidence interval, 0.96 – 1), 0.990 (95% confidence interval, 0.97 – 1), and 0.978 (95% confidence interval, 0.95 – 1), respectively. The average probabilities for identifying brain cancer, liver cancer and control samples using DMR+DHMR-based models were 0.72, 0.75 and 0.76, respectively (Fig. 4e). Furthermore, the models also predicted early stage and late stage of liver cancer samples in the validation cohort equally well (Supplemental Fig. 6). Finally, two other machine learning models (random forest and DNN) using both DMRs and DHMRs as inputs were also robustly better than models using DMRs or DHMRs alone (Supplementary Fig. 5a–f). Together, these studies indicate that the sscf-MeDIP-Seq method developed here provides a unique way to analyze both cfDNA DMRs and DHMRs, the latter of which have not been used for tumor detection.

## Evaluate the sensitivity of the sscf-MeDIP-Seq method

The amount of cfDNA in plasma differs from sample to sample, with early-stage tumors in general releasing less circulating tumor DNA into

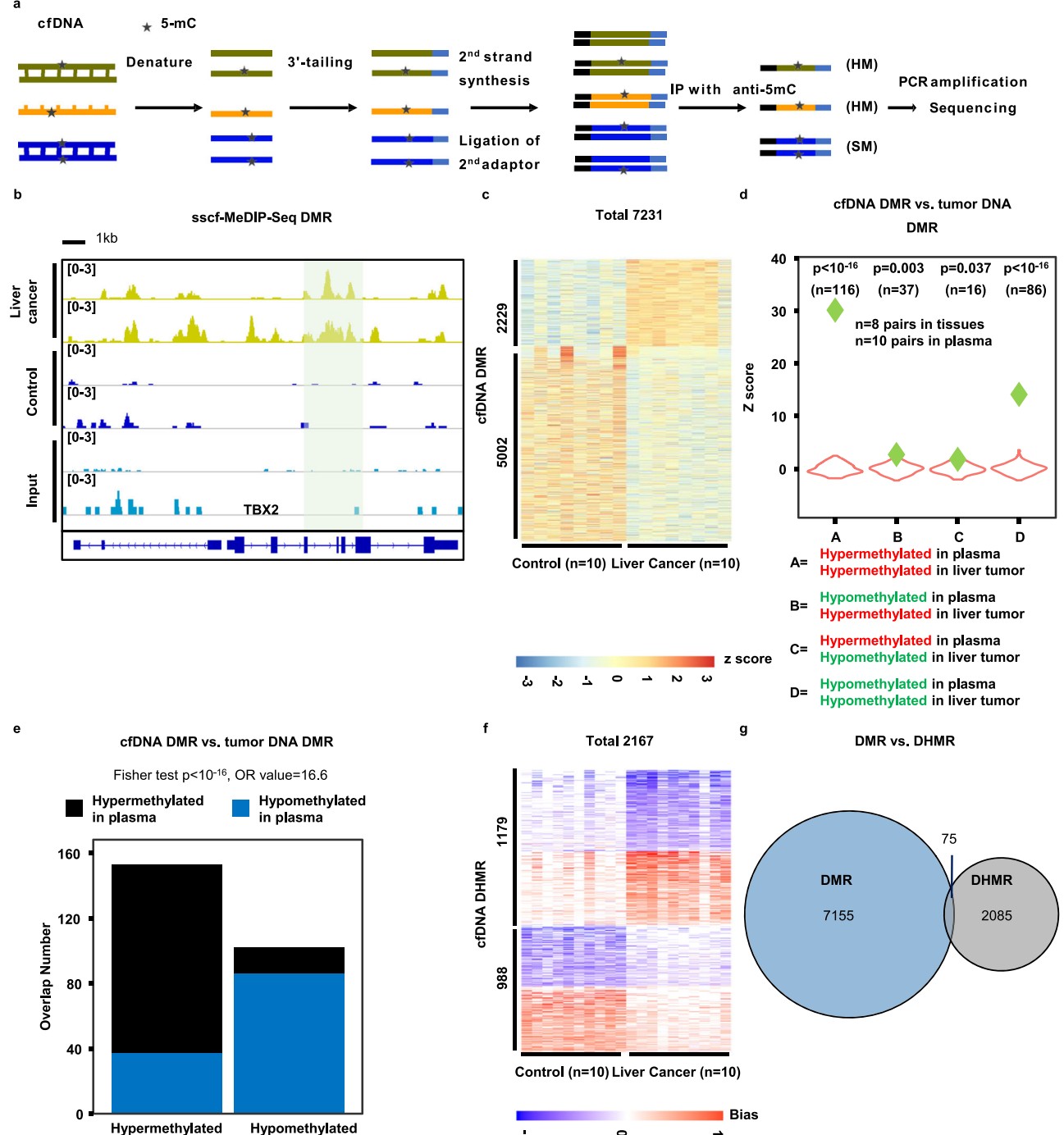

**Fig. 3 | A sscf-MeDIP-Seq procedure to analyze methylomes of plasma cell free DNA (cfDNA). a** An outline of the sscf-MeDIP-Seq method for analyzing cfDNA methylomes. SM symmetric DNA methylation, HM hemi-methylation. Please note that DNA methylation on single-stranded (ss) DNA would be regarded as HMR. Based on 8 input samples, the number of HMRs that arose from ssDNAs was likely to be small. **b** A snapshot of cfDNA DMR at the *TBX2* gene locus. The shaded area highlight DMRs identified using 10 cfDNA samples from liver tumor patients compared to 10 cfDNA samples from controls, with only two samples from each group shown. Also shown are sequence reads of two input samples in which methylated DNA immunoprecipitation was also not performed. **c** Heatmap of cfDNA DMRs from 10 plasma samples of liver cancer and 10 plasma samples of non-tumor controls. The Z score, shown in color, represents log$_2$ (RPKM) of sscf-MeDIP-Seq signals. Violin plot (**d**) and bar plot (**e**) showing overlaps between liver tumor cfDNA DMRs identified by sscf-MeDIP-Seq and liver tumor DNA DMRs identified in this study using ssg-MeDIP-seq. Violin plots represent the random distribution of overlaps from 100 permutations and *P* values were computed using random permutation distribution in a one-side way. Green diamonds represent observed overlaps. Fisher test was used for statistical analysis of the bar plot in a two-side way. **f** Heatmap of plasma cfDNA DHMRs of 10 liver cancer samples and 10 controls. The hemi-methylation score, shown in color, was calculated using formula shown in Fig. 2a and represents HM levels. **g** Overlap of plasma cfDNA DMRs and cfDNA DHMRs of the same 10 liver cancer samples compared to 10 controls.

**Table 1 | Patient information including cancer type, sex and age of all 271 cfDNA samples used in this study**

| | Training cohort (N = 215) | Validation cohort (N = 56) |
|---|---|---|
| *Sex and age* | | |
| Male | 127 | 34 |
| Female | 87 | 22 |
| Unknown | 1 | 0 |
| *Age at diagnosis/ recruitment* | | |
| Young ( < 30 years) | 13 | 6 |
| Middle age (30 ~ 60 years) | 107 | 25 |
| Old age ( > 60 years) | 95 | 25 |
| Unknown | 0 | 0 |
| *Brain cancer* | 77 | 20 |
| *IDH* WT | 34 | 9 |
| *IDH* mutant | 43 | 11 |
| *Liver cancer* | 58 | 15 |
| *Controls* | 80 | 21 |

blood than late-stage tumors[43,44]. Therefore, we normally used 1/3-/1/2 of cfDNA purified from 1–1.5 mL plasma sample for sscf-MeDIP-Seq experiments. To test the amount of cfDNA needed for the generation of high quality sscf-MeDIP-Seq datasets for tumor prediction, we chose two cfDNA samples with high cfDNA concentration, one from individual with liver tumor and one with brain tumor, and then generated three sscf-MeDIP-Seq datasets using three different amounts of cfDNA. We also used the fraction of cfDNAs from each sample (3.5 ng (1/48), 10 ng (1/16), 24 ng (1/7 of the sample) for the brain cancer; and 3 ng (1/20), 7 ng (1/8) and 15 ng (1/4) for the liver cancer, Supplemental Fig. 7) instead of the exact cfDNA amount when we generated sscf-MeDIP-Seq libraries. We then applied the GLMnet models trained in Fig. 4 to predict these samples based on sscf-MeDIP-seq datasets generated from different amounts of input DNAs. The DMR + DHMR-based models could predict brain and liver cancer samples at all three concentrations (Supplemental Fig. 7a, d). In contrast, the DMR- and DHMR-based models could reliably predict brain or liver cancer based on sscf-MeDIP-seq datasets from two different amounts of cfDNAs (Supplemental Fig. 7b, c and e, f). These results are consistent with the idea that DMR+DHMR-based models will likely be more robust in predicting tumor types. Furthermore, these results indicate that in general a higher input cfDNA used for sscf-MeDIP-Seq yielded better quality of sscf-MeDIP-Seq datasets for prediction. Therefore, we generated all 271 sscf-MeDIP-Seq datasets using cfDNAs purified 300 μl to 500 μl of plasma samples, which are equivalent to 1/3-1/2 cfDNA purified from 1 – 1.5 mL plasma of the majority of samples analyzed in this study.

**Differentiate glioma subtypes by cfDNA methylomes**
We also tested whether cfDNA methylome analysis can be used to differentiate the subtypes of brain tumors. Of 77 cfDNA samples from brain tumor patients in the training cohort, 43 samples were from patients with *IDH* mutations and 34 with *IDH* wild type. To train brain tumor subtype models, we first separated the 77 brain tumors samples of the training cohort into *IDH* mutant (43 samples) and *IDH* wild type groups (34 samples) and followed the same procedures outlined above to train the GLMnet models using either DMRs or DHMRs as inputs. These brain subtype models were then combined with the three-class model (brain cancer, liver cancer and control) based on Bayes's theorem to expand the model for four samples groups (*IDH* WT and *IDH* mutant brain cancer, liver cancer, and control) (Fig. 5a). Using the four-sample class model, we calculated the prediction probability of each sample in the validation cohort. As shown in Fig. 5b, c, we could

identify *IDH* mutant and *IDH* wild type brain tumor subtypes accurately, with the DMR+DHMR-based models having the best performance (AUROC of 0.947 (95% confidence interval, 0.88 – 1) and 0.955 (95% confidence interval, 0.9 – 1) for *IDH* mutant and *IDH* WT, respectively). Finally, the average probabilities of *IDH* mutation gliomas, *IDH* wild type gliomas, liver cancer and control groups were 0.55, 0.40, 0.72 and 0.74, respectively (Fig. 5d). Together, these studies indicate that models using both DMRs and DHMRs as inputs could also be used identify glioma subtypes accurately.

**cfDNA DMRs are associated with genes whose gene expressions in tumor tissue samples predict patient survival**
Promoter and enhancer DNA methylation is associated with gene transcription[45,46]. To probe the potential relationship between cfDNA DMRs and gene expression in tumor samples, we first annotated each of the liver cancer specific 10,051 cfDNA DMRs, which were identified by comparing cfDNA methylomes of all 58 liver cancer samples in the training cohort to those from control and brain tumor samples in the training cohorts, to their closest genes and identified 1689 genes whose promoters were within 20 Kb of one of these DMRs. We then asked whether the expression of each of the 1689 genes in 371 liver tumor samples in the TCGA database was associated with patient survival (Fig. 6a). For instance, a hypo-methylated DMR at the *SOX14* gene locus specific for liver cancer compared to controls and brain tumor samples was identified (Supplementary Fig. 8a). Furthermore, high expression of *SOX14* in the 371 TCGA liver cancer dataset was associated with poor survival compared to lower expression (Supplementary Fig. 8b). Through this analysis, we found that of the 1689 genes with at least one liver cancer specific cfDNA DMR nearby, the expression of 150 genes in liver cancer tissues in the TCGA database was associated with patient survival. Of these 150 genes, 62 genes were associated with hyper-methylated cfDNA DMRs, whereas 88 genes were close to hypo-methylated cfDNA DMRs (Fig. 6b). Next, we asked whether the expression of these 150 genes could be used to cluster the 371 TCGA liver cancer patient samples using unsupervised clustering analysis and found that these 371 samples could be separated into two clusters. Interestingly, genes close to the hypo-methylated cfDNA DMRs are highly expressed in "Cluster 2" liver tumor samples compared to "Cluster 1" (Fig. 6c). In contrast, genes close to hyper-methylated cfDNA DMRs are highly expressed in "Cluster 1" patient samples. Importantly, patients in these two clusters showed dramatically different survival times, with the median survival of patients in Cluster 1 and Cluster 2 being ~80 and ~30 months, respectively (Fig. 6d).

We also applied the same approach and identified 37 genes with at least one brain tumor specific cfDNA DMR, and the expression of these genes in primary brain tumor tissue samples was associated with patient survival (Supplemental Fig. 8c, d). The expression of the 37 genes in tumor tissues could also separate 156 brain tumor samples from the TCGA database into two different clusters with patients in "Cluster 2" showing better survival than those in "Cluster 1" (Supplementary Fig. 8e–g). Interestingly, we noted that patient samples with *IDH* mutations were enriched in "Cluster 2" (Fisher test, OR = 6.2, p = 0.01). It is known that brain tumor patients with *IDH* mutations have a favorable outcome compared to glioma patients with wild type *IDH* gliomas[47]. Together, these studies indicate that some cfDNA DMRs for both liver and brain tumor patients are likely associated with changes in expression of nearby genes involved in tumorigenesis.

## Discussion
DNA cytosine methylation plays an important role in gene regulation, chromatin maintenance and genomic stability[25]. Aberrant DNA methylation occurs in a variety of cancers. Therefore, DNA methylomes in cancer tissues have been used for tumor classification and detection. In this study, we developed ssg-MeDIP-seq procedures for

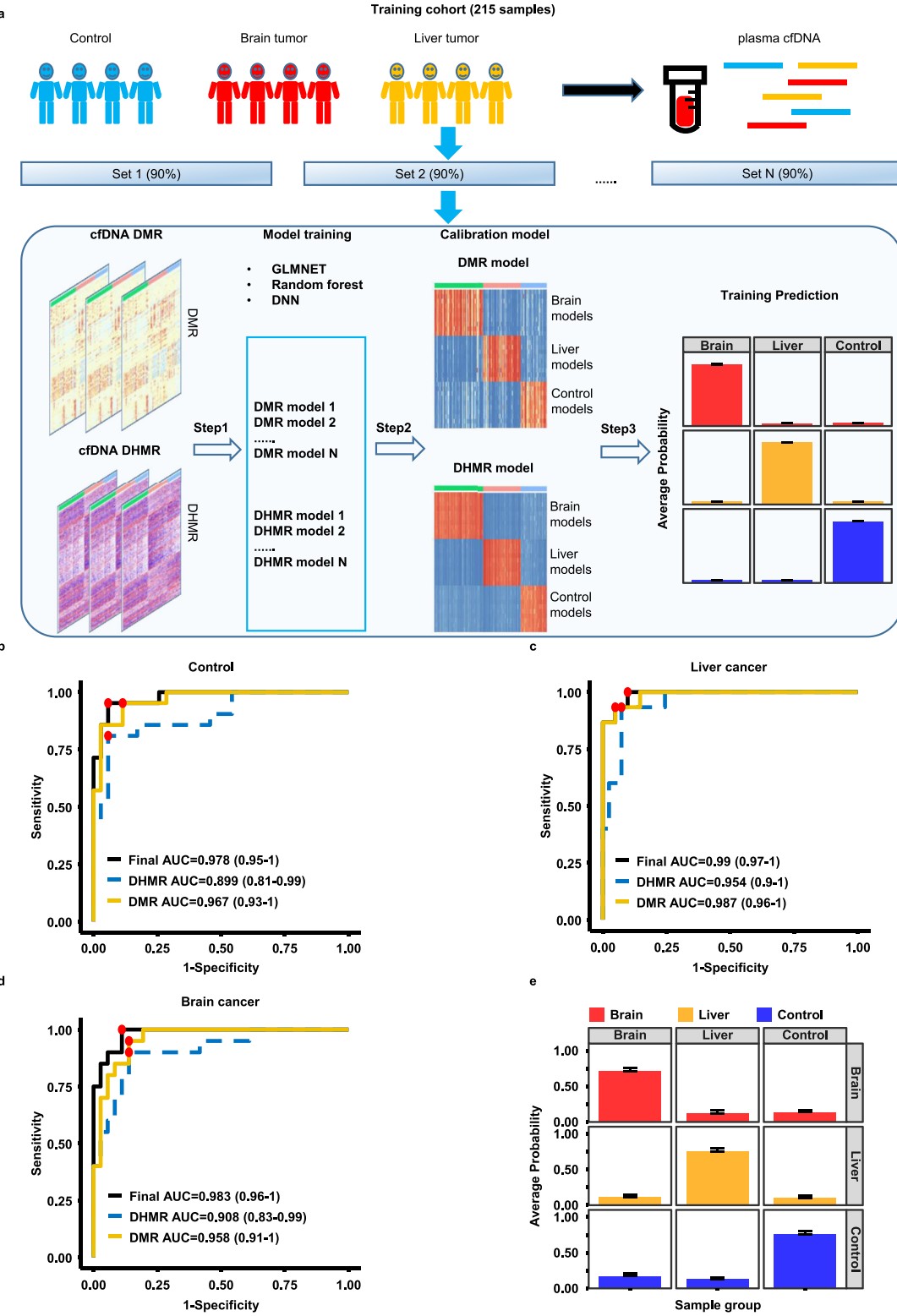

**Fig. 4 | Multi-cancer detection using DMRs and DHMRs and machine learning models. a** A workflow of machine learning model training. Methylomes of 271 cfDNA samples from three groups of individuals (controls, brain and liver cancer patients) were analyzed using sscf-MeDIP-Seq. 215 sscf-MeDIP-seq datasets (80%) were used as the training cohort and the remaining 56 (20%) samples as the independent validation cohort. The training cohort was used for DMR and DHMR selection and training of machine learning models, results in 10 models for each sample group using DMRs or DHMRs as the input for the training. The DMR- and DHMR- based models were then further unified to build a final calibration model. The validation cohort was then evaluated using models trained with DMRs, DHMRs and DMRs+DHMRs as inputs. Evaluation of model performances for the prediction of control (**b**), liver tumor (**c**) and brain tumor (**d**) cfDNA samples in the validation cohort using models trained with DMRs, DHMRs, or DMRs+DHMRs. The best sensitivity and specificity point for each prediction are marked in red dot. The 95% confidence interval of AUC for each model is labeled in parenthesis. **e** The average prediction probability of each group of samples using models trained with DMRs +DHMRs. Each column represents the group of validation samples, with each row representing model predictions. Bar plots are presented as mean value+standard error. Red, yellow and blue bars represent probability of samples being from 20 brain cancer, 15 liver cancer, and 21 healthy controls, respectively.

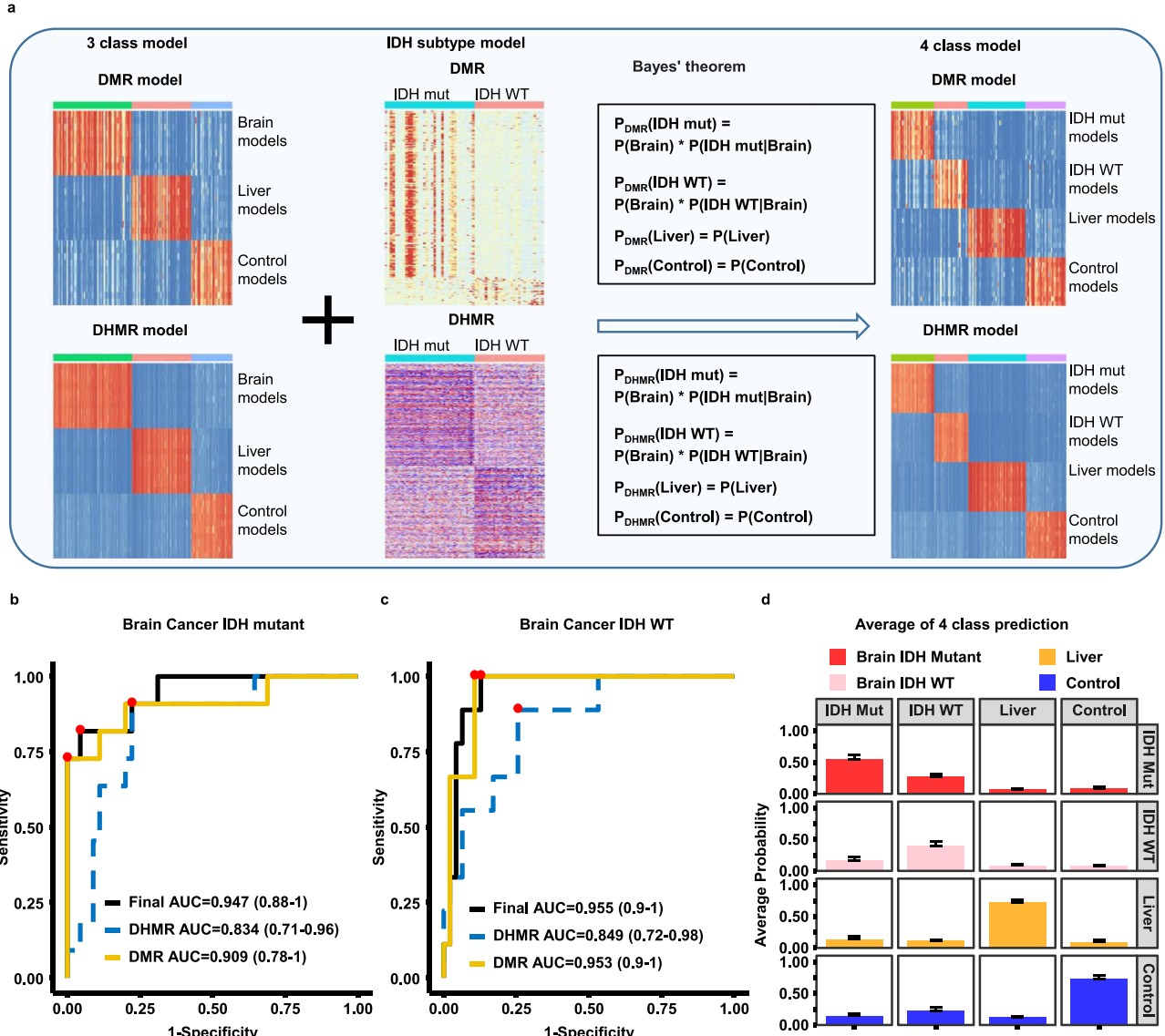

**Fig. 5 | Predicting brain tumor subtypes using sscf-MeDIP-Seq datasets. a** A workflow for building brain tumor subtype models. Models for the *IDH* WT and *IDH* mutant gliomas were first trained by DMRs and DHMRs identified using the training cohort samples and then combined with the three class models (controls, liver and brain tumor) based on the Bayes' theorem to derive models for predicting four sample groups: *IDH* WT and *IDH* mutant brain tumor, liver tumor and control samples. Evaluation of predicting *IDH* mutant (**b**) and *IDH* wild type (**c**) brain cancer samples in validation cohort using models trained with DMRs, DHMRs, DMRs +DHMRs. The best sensitivity and specificity point are labeled as red dots on the curve. The 95% confidence interval of AUC for each model is labeled in parenthesis. **d** The average prediction probability of each group of samples based on DMR +DHMR models. Each column represents the sample groups in the validation cohort, with each row representing model predictions. Bar plots are presented as mean value + standard error. Red, pink, yellow and blue bars represent probability of samples from 11 *IDH* mutant brain cancer, 9 *IDH* WT brain cancer, 15 liver cancer, and 21 controls, respectively.

analyzing genomic DNA methylomes as well as sscf-MeDIP-Seq for plasma cfDNA methylomes. These markedly simplified MeDIP-seq procedures greatly reduce the amount of DNA and time needed for the generation of MeDIP-seq datasets. Importantly, these methods allow us to analyze both symmetric DNA methylation as well as hemi-methylation, a recently-described epigenetic mark that has not been used for tumor detection[27,28]. Below, we discuss the implication of our findings in tumor detection and tumor classification.

To optimize and simplify MeDIP-seq procedures for analyzing DNA methylomes of genomic DNA isolated from normal or tumor tissues in a strand-specific manner, we utilized pA-Tn5 loaded with one adaptor, which tagments dsDNA into small fragments. Furthermore, because pA-Tn5 attaches the adaptor only to the 5' end of each strand covalently, another adaptor is used to replace the first adaptor at the 3' end following tagmentation, which makes it possible to detect DNA

methylation at both Watson and Crick strands separately. In this way, we do not need the sonication step for shearing DNA into small fragments, which is the first step in previously published MeDIP-seq procedures[48]. Furthermore, because of tagmentation, MeDIP-seq libraries are generated through a simple PCR step. These modifications allow us to generate high-quality MeDIP-seq libraries from 100 ng tumor DNA in less than two days. Importantly, ssg-MeDIP-Seq can measure both DNA methylation density and hemi-methylated regions at the same time.

DNA hemi-methylation was considered previously as the replication intermediate that will become fully methylated following DNA replication. Recently, it has been shown that about 10% of the ~3 million CpGs sites are hemi-methylated and are heritable, indicating that DNA hemi-methylation is a epigenetic mark[27,28]. We identified 190,106 and 228,575 HMR regions in 8 liver tumor samples and 8 adjacent controls tissues, respectively, which are about 10% of the ~2 M

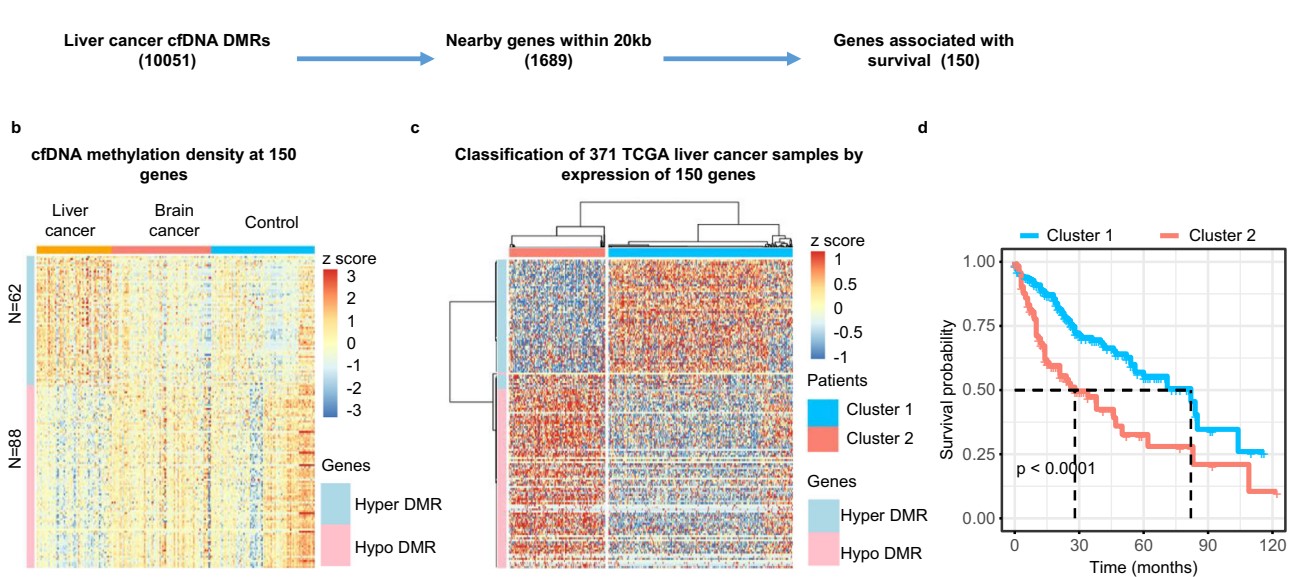

**Fig. 6 | Classification of liver cancer samples and prediction of patient survival based on the expression of genes in TCGA liver tumor tissues with a nearby liver-tumor specific plasma cfDNA DMR. a** An outline to identify genes having at least one liver cancer cfDNA DMR within 20Kb from their promoters and whose expression in TCGA liver tumor tissue samples being associated with patient survival. **b** sscf-MeDIP-seq density at DMRs close to the 150 genes with at least one cfDNA DMR nearby. The z-score, represented by color, is $\log_2$ (RPKM) of sscf-MeDIP-Seq signals. A "Hyper DMR" refers to a gene with at least one hyper-methylated cfDNA DMR nearby, A "Hypo DMR" is defined as a gene with a hypo-methylated cfDNA DMR nearby. **c** Classification of 371 liver tumors in the TCGA-LIHC cohort based on expression of the 150 marker genes identified above. Patients are classified into two clusters. The color represents the z-score of $\log_2$ (RPKM) of RNA-seq signals of 150 genes in 371 liver cancer samples. **d** Kaplan–Meier survival analysis of 371 liver cancer patients separated into two clusters as in (**c**). *P* value is calculated by log rank test. Note that the software did not provide the exact *p* value when it is smaller than 0.0001.

potential methylated blocks[38] used for analysis. We also identified 6864 differentially hemi-methylated regions (DHMRs) by comparing HMRs in liver tumor samples to those of adjacent non-tumor tissues. Importantly, we found that the majority of liver tumor DNA DHMRs do not overlap with DMRs in the same samples. The little overlap is likely due to the fact that DNA methylation density at liver tumor DMRs is altered equally at both strands, whereas DNA methylation density at DHMRs is altered markedly at one strand compared to the control samples. Together, these studies provide additional data to support the idea that HMRs are epigenetic marks and that DHMRs could be used as independent biomarkers.

Interestingly, we found that liver tumor DNA DHMRs are close to genes involved in metabolism. Previously, it has been shown that a fraction of HMRs is associated with the CTCF sites[27,28], and further-more, hemi-methylation at different strands has opposite effects on the binding of CTCF[29]. CTCF is important for genome organization. Therefore, it would be interesting to determine how DHMRs regulate gene expression of these genes involved in metabolism in liver cancer samples, and whether DHMRs in other tumor types are also close to genes involved in metabolism.

## Tumor detection by methylation and hemi-methylation of plasma cell free DNA

In contrast to large dsDNA fragments for genomic DNA, cfDNAs are a mixture of dsDNA and ssDNA and damaged DNA, which are similar to ancient DNA samples. Therefore, we used our previous experience on the generation of sequencing libraries from ssDNA to optimize and developed sscf-MeDIP-Seq procedures[49]. Previously, it has been shown that single-stranded DNA library preparation for next generation sequencing will retain the information provided by dsDNA preparation methods, and at the same time analyze ssDNA molecules that are not included for analysis by dsDNA preparation methods[36]. Therefore, these modifications allowed us to include double stranded, single-

stranded and damaged cfDNA for methylation analysis, and as such we could generate MeDIP-seq libraries from cfDNA purified from 300 – 500 µl plasma samples. Importantly, the sscf-MeDIP-Seq method can measure cfDNA methylation on Watson and Crick strands separately, which makes it possible to analyze cfDNA hemi-methylation. Like liver tumor DNA DHMRs, most liver cancer cfDNA DHMRs do not overlap with cfDNA DMRs. These results indicate that liver cancer cfDNA DHMRs are independent biomarkers. Indeed, by analyzing cfDNA methylomes of 271 plasma samples from three groups of individuals, we found that machine-learning models trained with DMRs or DHMRs alone show good performance predicting sample groups in the validation cohort, and furthermore, models trained with DMRs+DHMRs show a slight improvement compared to the models trained with DMRs or DHMRs alone. The slight improvement is likely due to the fact that DMR- or DHMR-based models already show great performance, which makes it challenging to improve further with the sample size we had in the validation cohort. The AUROC is 0.978, 0.990, and 0.983 in predicting control, liver and brain cancer samples vs other in the validation cohort, respectively, for models trained with DMRs+DHMRs in our study. This performance is similar to, if not better than, what has been reported by analysis of cfDNA methylomes using other methods for tumor detection. For instance, it has been shown that AUROC for predicting lung cancer, acute myeloid leukae-mia, pancreatic ductal adenocarcinoma and controls in the validation cohorts are 0. 971, 0.980, 0.918 and 0.969, respectively, through analysis of methylomes of cfDNA by an improved MeDIP-Seq[20]. In large cohort of studies on over 50 cancer types, the specificity for all cancer types is about 99.5% with 95% confidence intervals by analyzing DNA methylation using bisulfite sequencing[21,22]. In all these studies, DHMRs were not used for analysis and tumor detection.

In clinical settings, one could envision at least two benefits for tumor detection utilizing both DMRs and DHMRs. First, the slight improvement by the DMR+DHMR models will have a real benefit when

a large number of samples is analyzed. Second and importantly, by evaluating samples using three different models (DMRs, DHMRs and DMRs+DHMRs), we could envision to reduce false positives during cancer screening, a major issue for early tumor detection using current assays. Because we could predict the same sample three times independently, we could in principle flag the sample for additional tests if predictions from three models show discordance. On the other hand, if the prediction from three models show concordance, this would increase confidence in the prediction. Therefore, we anticipate that tumor prediction using different models trained with independent features will reduce the false positive rate and increase prediction accuracy. In the future, it would be interesting to test these ideas using a large cohort of samples and in clinical settings.

## Methods

This research project was approved by the Columbia University Institutional Review Board.

### Biospecimens

Hepatocellular cancer patients' samples were from an IRB-approved, hospital-based prospective study conducted at Columbia University Irving Medical Center (CUIMC) that recruited liver cancer patients (>18 years older) from Oct. 2008 to July 2014. Brain cancer patients' samples were collected as part of an IRB-approved protocol to collect, bank and distribute de-identified samples from brain tumor patients at CUIMC. Subjects without cancer were recruited from advertisements around CUIMC also with IRB approval. All subjects provided blood samples which were rapidly centrifuged to obtain plasma which was aliquoted and frozen at $-80\,°C$ until use. Basic epidemiologic variables were obtained by a structured questionnaire while clinical information on patients was obtained from medical records. Written informed consent was obtained from all participants. We did not perform analysis based on sex and gender in this study because we aimed developing models for tumor detection in a sex and gender independent way. However, because there are likely sex-and gender-specific DNA methylation patterns, we attempted to include equal number of female and male samples in this study. Sex and gender was based on self-report.

### Protein, antibody and reagents

Purification of pA-Tn5 and pA-Tn5-oligo complex assembly used for analysis of methylation of tumor DNA were as described previously[50,51]. Antibodies against 5-mC (33D3) were purchased from Diagenode (C15200081). Mouse IgG used to bridge antibodies against 5-mC and pA-TN5 was purchased from Active Motif (53017), and tRNA was purchased from Sigma (R1753).

### Preparation of genomic DNA

Genomic DNA was extracted from frozen tumor and adjacent tissues by standard proteinase K and RNase treatment followed by phenol and chloroform extraction.

Tagmentation of genomic DNA was performed as previously described with minor modifications[50,51]. In brief, 100 ng of DNA and 1.5 μl of pA-Tn5-AA complex were mixed in the Tagmentation buffer (5 mM TAPS-NaOH pH8.5, 5 mM MaCl₂, 10% DMF), and were incubated in 37 °C with gentle shaking for 30 min. DNA was then purified by CHIP DNA clean kit (Zymo 11 − 379 C), and oligo replacement and GAP repair followed the same procedures as described[50,51]. The processed DNA was then subjected to immunoprecipitation using antibodies against 5-mC described below.

### Methylated DNA immunoprecipitation (MeDIP) and library preparation

The processed DNAs were diluted to 200 μl with the binding buffer (50 mM Tris pH 8, 350 mM NaCl, 0.05%Triton X-100, 1 mM EDTA), heated to 98 °C for 10 min, then cooled on ice immediately for 5 min.

5 μg tRNA (R1753 sigma) and 0.6 μg anti-5-mC monoclonal antibody 33D3 (C15200081) were added to the mixture, rotated at 4 °C for 1 h. After addition of 1 μl of bridge antibody (Active Motif 53017) and 10 μl pre-washed Protein G beads (Invitrogen 10004D), the reaction mixtures were incubated at 4 °C for 16 h. After incubation, protein G beads were washed twice with the binding buffer, twice with wash buffer (50 mM Tris pH8, 140 mM NaCl, 0.05% Triton X-100, 1 mM EDTA) and twice with TE buffer. DNA on the beads was eluted twice at 65 °C for 15 min with 15 μl Elution buffer (10 mM Tris-HCl, pH8.0, 10 mM EDTA, 150 mM NaCl, 5 mM DTT, 1% SDS). Eluted DNAs were then combined and purified with CHIP DNA Clean & Concentrator (Zymo 11-379 C). The purified DNAs were eluted in 20 μl low EDTA TE (Swift 90296). For the genomic DNAs, Illumina Nextera Dual Index primers were used for library amplification. Briefly, PCR reactions consisting of 20 μl eluted DNA, 1.5 μl 10 mM N7 primer, 1.5 μl 10 mM N5 primer, and 23 μl 2X PCR master mix (NEB 0541 S) were assembled for library preparation.

Plasma cell free DNA was purified by following the procedures of QIAGEN kits (QIAamp MinElute ccfDNA Mini Kit). To perform methylated DNA immunoprecipitation of plasma cfDNA samples, 1 S Plus Set Indexing kits (Swift 16024) were used for sample indexing. Briefly, reaction mixtures consisting of 20 μl eluted DNA, 5 μl R1, 25 μl 2X PCR master were assembled in PCR tubes and used for PCR amplification (98 °C 1 min: 98 °C 10 s, 63 °C 20 s, 10-11 cycles: 72 °C, 1 min). After PCR amplification, the reaction mixtures were mixed with 25 μl AMPure XP beads (Beckman A63880) for 5 min at room temperature. The supernatant was then transferred to a new tube with 25 μl of AMPure XP beads. After a 10 min incubation at room temperature, the DNA on beads were washed twice with 200 μl 80% ethanol, and eluted with 14 μl low EDTA TE.

### MeDIP-Seq data analysis

MeDIP-Seq libraries were sequenced using a paired-end method on Illumina Nextseq 500/550 or NOVA-seq platforms Adaptor sequences of all raw reads were removed by Cutadapt[52] and reads <10 nt were removed. Reads passed through these cleanup steps were then mapped to the human reference genome (hg19) by Bowtie2[53]. Duplicate reads were removed using Sambamba software[54]. Read coverage in a bin of 1 bp was calculated from filtered bam files by deepTools2[55] and then normalized with total number of filtered reads into reads per million (RPM).

Protein coding gene annotation was downloaded from GENCODE (v28)[56] and the CpG islands annotation was downloaded from UCSC Table Browser[57]. Protein coding genes were then classified into genes with and without CpG islands based on the overlap with their promoters ([− 3 kb, 3 kb] surrounding TSS). Normalized reads density (RPM) of MeDIP-Seq was used to calculate from transcription start sites (TSS) to transcription termination sites (TTS) for each class of genes respectively by deepTools2[55].

### Differentially methylated region (DMR) identification for genomic DNA and plasma cfDNA

Recently, it has been shown that 2,002,724 blocks each with at least four CpG dinucleotides can monitor DNA methylation from 205 tissues across multiple conditions[38]. Therefore, we used 2,002,724 blocks with at least four CpG dinucleotides to identify DMRs and DHMRs in our study. To identify DMR of genomic DNA of liver tumor tissues and adjacent non-tumor tissues, we compared ssg-MeDIP-seq datasets from eight liver tumor samples to each corresponding adjacent non-tumor tissues, and identified 11,930 hyper-methylated DMRs and 12,974 hypo-methylated DMRs by QSEA[58] with a cutoff of $p < 0.01$ and |log₂(fold change)|>1. For comparison, TCGA LIHC[59] methylation datasets of Illumina 450 K CpG array were downloaded. Methylation level β values of 50 liver tumor samples were compared with their corresponding adjacent samples, and 10,362 hypermethylated DMRs and 46,969 hypomethylated DMRs were identified using the T-test with a cutoff of

Bonferroni adjusted $p < 0.05$. To estimate the significance of overlaps between liver cancer DNA DMRs identified in this study and DMRs identified using TCGA datasets, 100 permutations were performed by bedtools[60] with the command "bedtools shuffle −incl regulation.bed". 5207 concordantly hyper-methylated and 1472 concordantly hypo-methylated DMRs between the liver cancer DMRs found in this study and TCGA liver tumor samples using 450 K methylation arrays were identified. The observed number of overlapping DMR (5207 and 1472) was compared with the null hypothesis distribution generated from corresponding 100 permutations and standard normalized to Z score ($P$ value is calculated by the null distribution in a one-sided way). Both of the concordant hyper-methylated and hypo-methylated DMRs showed significantly ($p = 0$ for 5207 hyper-methylated DMRs and $p = 0$ for 1472 hypo-methylated DMRs, respectively) higher enrichment than random permutation distributions (Fig. 1d).

To identify cfDNA DMRs, we compared cfDNAs sscf-MeDIP-Seq datasets from 10 liver cancer patients with those from 10 control individuals without cancer using QSEA[58] with a cutoff of $p < 0.01$ and $|\log_2 (\text{fold change})| > 1$. To evaluate whether these cfDNA DMRs were also found in liver tumor DNA DMRs, we first compared the overlap between cfDNA DMRs with tumor DNA DMRs identified using ssg-MeDIP-seq in this study. Similarly, discordantly methylated regions (hyper-methylated in one setting vs hypo-methylated in another setting) between cfDNA DMR and DMRs on tumor DNAs from this study were also compared. Subsequently, the observed numbers of concordantly and discordantly overlapped DMRs were compared with the null distributions generated from the corresponding 100 random permutations. The observed numbers of overlapping DMRs were normalized to Z score by null distributions and $p$ values were calculated in a one-sided way to evaluate the significance of overlap.

### Differentially hemi-methylated region (DHMR) identification for genomic DNA and plasma cfDNA

The same 2,002,724 blocks from Loyfer et al.[38] were also used to identify hemi-methylated regions (HMRs) and differentially hemi-methylated regions (DHMR). Briefly, hemi-methylation level (HM) at each block was calculated as "bias" $= \frac{Watson - Crick}{Watson + Crick}$, which ranges from −1 to 1. Watson and Crick represent ssg-MeDIP-Seq or sscf-MeDIP-Seq sequence reads of Watson and Crick strand, respectively. HMRs were identified using a cutoff of the absolute bias greater than 0.3[27]. To identify liver tumor DNA DHMR, eight ssg-MeDIP-Seq datasets from liver tumor were compared to their adjacent non-tumor tissues (Fig. 2d). To identify cfDNA DHMRs for detailed analysis, cfDNA from 10 individuals with liver tumor and 10 control individuals without liver cancer were compared (Fig. 3f). We used T-test on each DNA methylation block with the cutoffs of $p < 0.01$, and difference in bias between samples > 0.3 and minimum bias at each block > 0.3. Finally, based on the impact of RPM at each block on the number of HMRs for the input samples, we included (RPM > 1 at each block) as the additional cutoff based on analysis of 10 input samples whose libraries were prepared using the same procedures as ssg-MeDIP-Seq or sscf-MeDIP-Seq for sequencing except that the methylated DNA immunoprecipitation step was not performed. If sequence read at a DNA methylation block is less than 1 (RPM), the block will be labeled as NA in the HMR score sheets. In the model training, we used the score "0" to represent the block. Of note, the NA score for the block could arise from 1) hypo-methylated regions, and/or 2) low sequence depth of the sample. Furthermore, a perfectly symmetric methylated regions is also labeled as "0" for model training. These imputations may contribute to a slight lower performance for the DHMR model than DMR models.

### Machine learning models trained using DMRs, DHMRs, or DMRs + DHMRs

To detect and classify tumors by cfDNA methylomes, we used a regularized regression model of Lasso and Elastic-Net Regularized Generalized Linear Model (GLMnet) as the final model, and also tested two other machine learning models, Deep Neural Network (DNN) and Random Forest (RF). To evaluate the performance of machine learning models, the receiver operating characteristic (ROC) curve was plotted in sensitivity against (1-specificity) for each class, where sensitivity = (true positives/total positives) and specificity = (true negatives / total negatives). The area under ROC curve (AUROC) were calculated for comparison by "pROC" package in R[61].

For GLMnet model, the elastic net penalty is controlled by α, bridging the gap between lasso regression (α = 1) and ridge regression (α = 0). In our study, α and λ were tuned over a grid of values to optimize the model from 0 to 1 in increments of 0.1, and the family function was set to "glmnet" for regression. For random forest model, the number of subset features for random selection was tuned over a grid of values by from 2 to the squared root of total number of features (DMRs or DHMRs), and 1000 trees were generated in each round. Model training was performed using 10-fold cross-validation and applied by "caret" package in R[62]. For the Deep Neural Network, our models consisted of an input layer, an output layer of three nodes for the predicted probability of each of the three types, and two hidden layers with 64 and 32 nodes. To process the input signals, the activation function for hidden layers were linear functions; and the output layer was the "softmax" function for multinomial classification. The L2 penalty was set to 0.1 for regularization of the hidden layers to reduce the risk of overfitting. We applied the DNN models by "Keras" package in R[63].

### cfDNA DMRs are associated with patient survival

We annotated cancer specific DMRs to their closest genes within 20 kb. Then we downloaded the RNA expression in RSEM value (RNA-Seq by Expectation Maximization) and patient's clinical data from TCGA-LIHC project (371 liver cancer patient samples) and TCGA-GBM project (156 brain cancer patients)[59,64]. Based on the median RNA expression of each of these nearby genes in these cohorts of patient samples, we separated liver or brain cancer samples into two groups, high and low expression. The cox proportional hazards model was performed for each of the nearby genes to evaluate hazard ratio on patients' survival[65]. The genes whose expression in these cancer samples is associated with patient survival were chosen for further analysis.

### Statistics & reproducibility

Statistical methods used for analysis were described in Figure legends. No statistical method was used to predetermine sample size. Furthermore, no data were excluded from the analyse. The experiments were not randomized, and the Investigators were not blinded to allocation during experiments and outcome assessment.

### Reporting summary

Further information on research design is available in the Nature Portfolio Reporting Summary linked to this article.

## Data availability

All sequencing and full datasets were deposited at dbGAP, with summary information being found at https://www.ncbi.nlm.nih.gov/projects/gap/cgi-bin/study.cgi?study_id=phs003462). DbGAP study accession: phs003462.v1.p1. To access these datasets, one first need to apply for access to the DbGAP database from NIH. Once granted, please contact Dr. Zhang at zz2401@cumc.columbia.edu requesting access datasets generated in this study. No restrictions will be placed on non-profit organizations. We will approve the access as soon as we can, normally less than 6 business days. Source data are provided with this paper.

## Code availability

The custom code was uploaded to github: https://github.com/clouds-drift/plasma_MCD.

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

## Acknowledgements

This work is supported in part by a pilot grant from Herbert Irving Comprehensive Cancer Center at Columbia University supported by P30CA013696, by NIH grants (R35 GM118015, R01 NS132344-01 and R01 CA277605-01A1 (ZZ), and R35 CA253126 (RR)). Biospecimens were processed and stored in the Biomarkers Laboratory supported by P30 ES009089 and P30 CA013696. We thank Drs. Wei Li for discussion and suggestions for methylomes analysis and Dr. Zhonghua Liu for discussion of machine learning models. The brain tumor study is supported by William Rhodes and Louise Tilzer Rhodes Center for Glioblastoma.
We thank the Herbert Irving Comprehensive Cancer Center Database Shared Resource as genome sequencing for providing (clinical data, biospecimens, or both) for the facilitation of this project.

## Author contributions

H.Z. and Z.Z. designed MeDIP-Seq methods. H.Z. developed MeDIP-Seq methods and performed all MeDIP-Seq samples used in this study. X.H. performed all bioinformatic analysis of this study, with the guidance of R. R.H.W., R.M.S., and Z.Z. secured the initial funding for this project, and H.W. and R.M.S. provided plasma samples of liver cancer and healthy individuals as well as genomic DNA samples for liver cancer and adjacent regions with the help from J.M.G., J. F., C.K., J.B. and P.C. provided plasma samples from gliomas and discussions. X.H. and Z.Z. wrote the manuscript with the help of H.Z. All authors read and edit the manuscript.

## Competing interests

A patent application describing the MeDIP-Seq procedures for analysis of DNA methylation of genomic DNA and cfDNA for tumor detection was filed by the Columbia University with Z.Z., H.Z., H.W., and R.M.S. listed as inventors. U.S. Provisional Application PCT/US2022/070998. Title: methods to analyze DNA methylomes in tumor and plasma cell free DNA. All other authors declare no conflict of interest.
