## [Peer Review File · Nature Communications]

Tumor detection by analysis of both symmetric- and hemi-methylation of plasma cell-free DNAREVIEWER COMMENTS

Reviewer #1 (Remarks to the Author): expertise in DNA methylation bioinformatics

In "Sensitive and accurate tumor detection by methylation and hemi-methylation of plasma cell-free DNA", Hua et al. describe strand-specific MeDIP-Seq for analyzing DNA methylation in a strand-specific manner in cells ("genomic DNA") and plasma samples. They apply their methods to liver tumors and adjacent controls, as well as cell-free DNA from individuals without cancer and individuals with liver or brain cancer. They define sites of differential methylation (DMR) as well as differential hemimethylation (DHMR) and find that genomic DNA-derived DMRs overlap with plasmid-derived DMRs. Surprisingly, they find that DMRs do not overlap with DHMRs, both in genomic and plasma samples. The authors construct a model to classify samples, and models that include DMR and DHMRs outperform models using only DMR or DHMRs, and can classify control, liver, and IDH+/- gliomas with reasonable accuracy. The authors identify DMRs and DHMRs that are located near genes whose expression are associated with particular gene functions and with patient survival. Overall, the authors present a compelling argument for the study of single-strand methylation and its utility in the context of cancer detection using cell-free DNA and their described assays.

Major comments:

1. Strand-specific methylation could arise from biases in 5-mC antibody (33D3), pA-Tn5, or strand dropout during sample collection or library preparation. The authors need to address these and other sources of biases and perhaps provide data from orthogonal methylation assays (e.g. targeted bisulfite sequencing) at DMRs and DHMRs to validate their findings.
2. The finding that DMRs do not overlap with DHMRs is surprising to me. Figure 2f shows that increase of single-strand methylation bias is enriched in CGIs which are generally unmethylated in normal samples. The figure suggests that DHMRs exist where unmethylated CGIs gain strand-specific methylation, but I would assume this would result in a DMR as well. Could the authors provide more intuition about how DHMRs arise without creating DMRs? For example, plots showing raw read counts on +/- strands at DMR and DHMR sites for all samples would be useful.
3. The authors test significance of overlapping DMRs using a permutation test, but these tests and accompanying figures don't give a sense for the raw overlapping rates. Raw numbers for overlaps are also not provided in the main text or supplementary data. Additionally, unexpected comparisons have a 'significant' p-value (e.g. Fig 3d class C p-0.003) decreasing the utility of this method. For Fig 1d, Supp 2d,e, and Fig 3d the overlaps should be measured and shown in another way (e.g. upset plot, overlapping barplot or venn diagram) that will increase the interpretability for readers.
4. DMR analysis was not performed on the plasma-titration-sensitivity samples – they were only used as inputs to previously-trained models (page 14, "Evaluate the sensitivity of the sscf-MeDIP-Seq method"). How many DMRs could be identified in the lower-coverage samples, and do they overlap with the DMRs from the larger-input samples? Do the DHMRs drop out before the DMRs with lower coverage?
5. Processed annotated data should be shared as supplemental information (e.g. values in Fig 2d, 3c, 3f, etc.)

Minor comments:

1. Page 4 "...whole genome DNA methylation is the best.." - the reference suggests that a subset of methylation can perform better than whole-genome DNA methylation. The authors should revise their summary of the reference.
2. Page 7 "we analyzed the DNA methylation profiles of liver cancer from TCGA" raw numbers of

overlaps between these sets should be reported.

3. Page 7 “we analyzed the DNA methylation profiles of liver cancer from TCGA” to increase the utility of single-strand methylation analysis the authors should test whether the methylation levels from TCGA were similar to methylation levels in the ss-MeDIP samples (and point out that 450K TCGA samples cannot report strand-specific methylation information).

4. Page 8 “we identified 260,055 and 325,866 HMRs in 8 liver tumor samples and their Adj-NT controls, respectively.” Is this the sum of all HMRs or an average per sample? The average per sample should be reported as well. This also applies to reporting of cfDNA DMRs (page 10) and elsewhere where aggregate numbers are reported.

5. Page 13 – “90% of the training cohort” and “in a balanced way” need to be defined – is this 90% of samples or 90% of features?

6. Page 14 – Sample selection wording is unclear “Briefly, we randomly chose two cfDNA samples, with each from individuals with liver and brain tumors.”

7. Page 15 – “Evaluate the sensitivity of the sscf-MeDIP-Seq method.” Were the machine learning outcomes improved if the only-DMR or only-DHMR models were used?

8. Page 17 – “Together, these results indicate that a significant fraction of liver cancer specific cfDNA DMRs and DHMRs identified in this study are likely associated with changes in the expression of nearby genes in tumor cells, which in turn may contribute to tumorigenesis.” It is unclear what significant fraction this sentence refers to. There is no significance test discussed in this section, and 78/4989 does not seem significant enough to make this claim. This sentence should be substantiated, softened or removed.

9. Page 23, DMRs were called in blocks in cfDNA samples, but how were DMRs called in 450K TCGA samples? Were they called as individual probes or somehow expanded to include multiple probes?

10. Page 25 “Machine learning models” What were the inputs to the models? I assume methylation levels at DMRs and bias at DHMRs, but this should be explicit.

11. Page 26 R scripts should be shared via github

12. Page 26 “cfDNA DMRs or DHMR are associated with patient survival” hazard ratio cutoff should be provided here

13. Page 33, Figure 1a – it could be useful to show an example of hemimethylation in the schematic here

14. Page 33, Figure 1b, page 7 text “By inspection of MeDIP-seq signals at the gene locus of TBX2, a gene known to be methylated in liver cancer, we identified a DMR specifically in tumors compared to Adj-NT samples” How was this highlighted DMR selected? It looks like there are several other DMRs in this region.

15. Page 33, Figure 1d – p-value for B is 0 – is this correct?

16. Page 33, Figure 1e – Does the LINE value extend beyond the left axis? The x axis should be increased to show the limit of this value.

17. Page 33, Figure 1e – N's should be shown for each category as enrichment values are hard to interpret (probably in a supplementary figure if not possible in the main figure)

18. Page 33, Figure 1e – x-axis “Z score enrichment” is not defined. Do the authors mean “Z score”?

19. Page 35, Figure 2b shows hemimethylation with a scale of -3 to 3. However, the formula

implies a range of -1 to 1.

20. Page 35, Figure 2f labels "Increased DHMR" could be changed to "Increased HM" to reflect that the region isn't increasing, but the hemimethylation is changing.

21. Page 42, Figure 1a – how were the IDH subtype model DMRs selected? The heatmap suggests that the DMRs were selected between a different number of samples (~80/20 instead of the 43/34 IDHmut/WT samples described in the text).

22. Page 42 Figure 5a the schematic of the application of Bayes' theorem is unclear – the boxes make it appear that the P(IDH mut) and P(IDH WT) apply to DMRs while P(Liver) and P(Control) apply to DHMRs.

23. Page 44 Figure 6d/g – perhaps x-axis label should be "Time in months" or "Time (months)"?

24. Figure S1 – Suggest changing labels to "Promoters with CGI" and "Promoters without CGI"

Reviewer #2 (Remarks to the Author): expertise in cell free DNA methylation methods

"Sensitive and accurate tumor detection by methylation and hemi-methylation of plasma cdl-free DNA" by Hua et al. is a well written article detailing utility of MeDIP-Seq methods. The performance of their classifier models to identify cancer and is promising but it appears that DHMRs do not contribute very much to the performance given especially that there are many more DHMR regions than DMRs.

INTRODUCTION

The statement that targeted bisulfite sequencing "requires up to 80ml of plasma" doesn't seem accurate. Assuming the authors are referring to Liu et al. which states "Up to 80 ml whole blood was collected from all participants as part of the research study; only two tubes of plasma were processed separately per participant." 1) that's 80ml of blood, not plasma and 2) only 2 tubes of blood were analyzed which would yield ~8-10ml of plasma.

METHODS/RESULTS

The authors' ssg-MeDip-Seq ligates different adapters to the Watson and Crick strands, allowing for strand-specific analysis. From the description provided and the diagram (Figure 1A) I don't see how the same result could not be achieved in silico by doing a standard library prep and then simply splitting the bam file by strand after alignment.

As the authors point out, allele specific methylation is an overlooked and potentially useful biomarker. It is possible to analyze strand specific methylation from bisulfite sequencing; the biscuit BS-Seq aligner, for example, has an allele specific methylation subroutine. With the data that is currently present it is difficult to evaluate the advantage of strand-specific methylation analysis using ss(g/cf)-MeDIP-Seq compared to bisulfite sequencing, as these comparisons are not presented.

The strand specific adaptor ligation shown in figure 1A is absent in figure 3A where they detail the cell-free DNA library prep (sscf-MeDIP-Seq). Is the strand specific adaptor ligation performed in sscf-MeDIP-Seq? Or in this case is strand specific methylation resolved in silico?

The authors state that their sscf-MeDIP-Seq library prep is superior to other cfDNA methods in part because it can recover single stranded and damaged DNA, which is particularly important for cfDNA applications. However, the authors do not compare their method with traditional MeDIP which begs the question: how much is gained by performing sscf-MeDIP-Seq versus previously

published methods and studies on cfDNA MeDIP-Seq (PMID: 35065650, 31471598, 31471598)? Does it improve sequencing output given identical input? Does it significantly improve signal detection? A direct comparison to previously published methods would strengthen the results.

Sequencing metrics that might be used to evaluate ss(g/cf)-MeDIP-Seq are absent (total reads, alignment rate, duplication rate etc.). It would also be useful to know the sequencing depth at each of the ~2M CpG clusters the authors evaluated.

The overlap between DMRs and DHMRs indicates that they are largely independent. However hemi-methylation would be ~50% methylated, if only considering beta value, and therefore hypomethylated compared to average genomic methylation levels; I would expect more overlap. If you reduce the stringency of the filtering criteria for DMR analysis do you see an increase in the overlap?

In the model training, the authors "selected the top 100 DMRs and 741 DHMRs" for model training/testing. How were cutoffs determined ($n = 100$ DMR & $n = 741$ DHMR)? Seems a bit lopsided in favor of the DHMRs, which do not seem to be adding much to the performance of the model. The authors state that the DMR+DHMR model performs better than either DMR or DHMR models alone but the DMR model has 100 features, no? What's the performance of the DMR model if you take the top 841 DMRs, so the total number of features is equal? Or if you were to train using the top X DMRs and top X DHMRs – this would allow for more direct comparison between the feature types.

Did you force the cluster breaks in the heatmaps in figure 6? C/F appear to have been forcefully split by row and column groups using the `row_split` and `column_split` options, assuming the authors are using the ComplexHeatmap R package. If so, this should be stated.

ROC curve AUCs should have confidence intervals either stated in the text or annotated on the figure, ideally both.

DISCUSSION

The authors state many advantages of their methods, please provide direct comparisons to previously published methods.

Reviewer #3 (Remarks to the Author): expert in machine learning cfDNA analysis

The author employed an enhanced MeDIP-Seq technique to examine DNA methylation patterns in liver cancer and brain cancer, underscoring the effectiveness of utilizing both DMRs and DHMRs for accurate cancer detection. Nevertheless, in previous studies, bioinformatics approaches for stranded methylation detection and hemi-methylation region identification from MeDIP-Seq data have already been well-established. Furthermore, both MeDIP-Seq sequencing technology and the utilization of the Tn5 enzyme for fragmenting and tagging double-stranded DNA in Next-Generation Sequencing (NGS) are well-established, mature techniques, emphasizing a notable lack of significant innovation. Hemi-methylated DNA typically has a propensity to either become fully methylated or tend towards demethylation. Varying DMR identification thresholds can be used to extract methylation change data in these regions. Moreover, the gold standard for methylation identification, WGBS (Whole Genome Bisulfite Sequencing), can differentiate between the positive and negative strands to acquire strand-specific methylation changes. This highlights a notable lack of impact.

My main detailed concerns regarding this manuscript are as follows:

1. Hemi-methylated regions represent a relatively small fraction of the genome, yet the number of DHMRs (Differentially Hemi-Methylated Regions) is significantly greater than the count of DMRs (Differentially Methylated Regions). This raises questions about whether the threshold set for DMR identification may have resulted in some regions that could potentially be identified as DMRs going

unnoticed, ultimately leading to a limited overlap between DHMRs and DMRs. This undermines the conclusion that the factors contributing to DHMRs are independent of those associated with DMRs.

2.MeDIP-seq exhibits a proclivity for interrogating genomic regions characterized by low CpG density, and fewer CpG sites are more susceptible to sequencing technology errors and random inaccuracies, leading to bias in identifying hemi-methylated regions.

3.The accuracy and sensitivity of cfDNA abnormal methylation in cancer detection have been previously reported in earlier studies. And the title is too generic and fails to highlight the main content and innovative aspects of the article.

4.External cohorts are needed to validate the diagnostic accuracy of DHMRs for lung and liver cancer, thereby preventing overfitting in model development.

5.In the clinical setting, blood tests for late-stage cancer patients are not meaningful. What is the distribution of cancer stages in your cohort? It is necessary to separately examine the diagnostic accuracy of early-stage DHMRs to demonstrate their practical significance in cancer diagnosis.

6.Raw data and code should be provided to ensure that data availability allows independent verification of results and increases the transparency of scientific research.

We thank all reviewers for their time to review this manuscript and for their insightful comments. We have performed additional analysis and experiments as well as editing to address each concern of all reviewers. Consequently, we felt that the revised manuscript has improved significantly.

Reviewer #1 (Remarks to the Author): expertise in DNA methylation bioinformatics

In “Sensitive and accurate tumor detection by methylation and hemi-methylation of plasma cell-free DNA”, Hua et al. describe strand-specific MeDIP-Seq for analyzing DNA methylation in a strand-specific manner in cells (“genomic DNA”) and plasma samples. They apply their methods to liver tumors and adjacent controls, as well as cell-free DNA from individuals without cancer and individuals with liver or brain cancer. They define sites of differential methylation (DMR) as well as differential hemimethylation (DHMR) and find that genomic DNA-derived DMRs overlap with plasmid-derived DMRs. Surprisingly, they find that DMRs do not overlap with DHMRs, both in genomic and plasma samples. The authors construct a model to classify samples, and models that include DMR and DHMRs outperform models using only DMR or DHMRs, and can classify control, liver, and IDH+/- gliomas with reasonable accuracy. The authors identify DMRs and DHMRs that are located near genes whose expression are associated with particular gene functions and with patient survival. Overall, the authors present a compelling argument for the study of single-strand methylation and its utility in the context of cancer detection using cell-free DNA and their described assays.

Response: We thank the reviewer for his/her time to review this manuscript and for the very positive and insightful comments. We have attempted to address each concern of the reviewer.

Reviewer #1:

Major comments:

1. Strand-specific methylation could arise from biases in 5-mC antibody (33D3), pA-Tn5, or strand dropout during sample collection or library preparation. The authors need to address these and other sources of biases and perhaps provide data from orthogonal methylation assays (e.g. targeted bisulfite sequencing) at DMRs and DHMRs to validate their findings.

Response: To address the reviewer’s concern whether the strand-specific methylation could arise from pA-Tn5 or strand dropout during sample collection or library preparation, we followed the same experimental procedures and prepared libraries using pA-Tn5 for genomic DNA of 2 liver samples without the methylated DNA immunoprecipitation step. We also followed the same procedures and prepared the libraries of 8 cfDNA samples (four control cfDNA samples and four liver cancer cfDNAs samples) without the methylated DNA immunoprecipitation step. We then used these input samples to assess the contributions of experimental procedures to the potential biases (hemi-methylation) at the ~2 M methylation blocks. Overall, the majority of blocks of the input samples did not show bias (Supplemental Fig. S2a, e), supporting the idea that the experimental procedures did not lead to the generation of strand-specific methylation bias. To minimize the contribution of sequence reads to false identification of hemi-methylated regions, we also tested the number of reads (RPM) at each block on the strand bias using these input samples. We found that the number of blocks showing bias decreased markedly using RPM>1 as the cut off compared to the cutoff of RPM>0.5. A

further increase of the cutoff to RPM>2 did not dramatically change the number of blocks showing bias based on these input samples (Supplemental Figure 2b, f). Therefore, we used the same cutoff and reanalyzed hemi-methylated regions of all samples using this cutoff. In this way, the bias generated by pA-Tn5 and library preparation procedures in MeDIP-Seq samples should be greatly minimized. Indeed, with this cutoff, we identified far fewer HMR regions and DHMRs for liver tumor samples and cfDNA samples than previously. Importantly, we found that machine learning models based on DHMRs performed significantly better than previous DHMR-models. For instance, the AUCs for control, liver and brain tumor samples were 0.761, 0.933 and 0.885, respectively, based on models trained with DHMRs identified previously (Figure 4 of previous version). The AUCs for control, liver and brain tumor samples were 0.899, 0.954 and 0.908, respectively, based on DHMRs identified in the revised manuscript (Figure 4).

We did not address whether 5-mC antibodies (33D) will generate bias for the following reasons. Using RPM>1 at each block as the cut-off, we found that only 10% of ~2M methylation blocks showed hemi-methylation (Figure 2c). It is unlikely that 33D antibodies will generate DNA hemi-methylation at these blocks only.

To further address this concern bioinformatically, we determined the percentage of hemi-methylated regions that overlapped with CTCF sites. We found that in both liver tumor DNA and their corresponding non-tumor control samples, about 1.2% HMRs overlapped with the CTCF sites in the genome. This is consistent with previous studies showing that about 0.8-1.4% of HMR overlapped with CTCF sites from two independent studies (Thomas et al, Nucleic Acid Research 2023, 51: 5997-6005, Table 1).

Reviewer #1:

2. The finding that DMRs do not overlap with DHMRs is surprising to me. Figure 2f shows that increase of single-strand methylation bias is enriched in CGIs which are generally unmethylated in normal samples. The figure suggests that DHMRs exist where unmethylated CGIs gain strand-specific methylation, but I would assume this would result in a DMR as well. Could the authors provide more intuition about how DHMRs arise without creating DMRs? For example, plots showing raw read counts on +/- strands at DMR and DHMR sites for all samples would be useful.

Response: We agree with the reviewer that it is surprising that a large fraction of DHMRs do not overlap with DMRs. As discussed above, with RPM>1 as the cut off, we found input samples showed bias or potential false positive HMRs at very few blocks. Using this cutoff, we found that while the number of DHMRs identified were reduced markedly compared to previous calculations without this cut off, about 2/3 liver tumor DNA DHMR (4474/6562) did not overlap with DMRs (Fig. 2e). Similar results were observed between cfDNA DHMRs and DMRs of 10 liver tumor samples compared to controls (Figure 3g). These results are consistent with the idea that DHMRs are novel epigenetic markers that can be inherited during cell division. We would like to point out that a recent study from the laboratory of Dr. Peter Jones, the pioneer of DNA methylation studies in cancer, found that hemi-methylation of different DNA strands at CTCF sites affect CTCF binding differently (Thomas et al, Nucleic Acid Research 2023, 51: 5997-6005). This study provides mechanistic insight into the potential function of hemi-methylation as well as additional rationale to use hemi-methylation as an independent biomarker.

To address the reviewer's comment directly, we calculated the normalized read counts of Watson and Crick strands at DMRs and DHMRs of 8 liver tumor DNA samples compared to their corresponding controls. We found that liver tumor DMRs in general showed alerted (increased or decreased) DNA methylation density at both Watson and Crick strands compared to the corresponding controls. In contrast, liver tumor DHMRs in general displayed altered DNA methylation density at only one strand compared to Adj-NT controls (Supplemental Figure 3). We would like to point out that we did not use raw read counts in this calculation because of different sequencing depth among these samples. Together, these results suggest that DMRs arise from changes in DNA methylation on both strands, whereas DHMRs arise from changes in DNA methylation at one strand, providing additional support to the idea that the majority of DHMRs is independent biomarkers from DMRs.

Reviewer #1:

3. The authors test significance of overlapping DMRs using a permutation test, but these tests and accompanying figures don't give a sense for the raw overlapping rates. Raw numbers for overlaps are also not provided in the main text or supplementary data. Additionally, unexpected comparisons have a 'significant' p-value (e.g. Fig 3d class C p-0.003) decreasing the utility of this method. For Fig 1d, Supp 2d,e, and Fig 3d the overlaps should be measured and shown in another way (e.g. upset plot, overlapping barplot or venn diagram) that will increase the interpretability for readers.

Response: We followed the reviewer's suggestion and included bar plots for all these figure panels (Figure 1e, Figure 3e and Supplemental Figure 4e). In fact, the bar plot results appeared to show more significant overlaps between concordant groups.

Reviewer #1:

4. DMR analysis was not performed on the plasma-titration-sensitivity samples – they were only used as inputs to previously-trained models (page 14, "Evaluate the sensitivity of the sscf-MeDIP-Seq method"). How many DMRs could be identified in the lower-coverage samples, and do they overlap with the DMRs from the larger-input samples? Do the DHMRs drop out before the DMRs with lower coverage?

Response: The reviewer raised several interesting questions. The purpose of testing different amounts of samples for the generation of sscf-MeDIP-Seq datasets was to determine whether sscf-MeDIP-Seq datasets generated using different amount of cfDNA samples would yield robust predictions by the trained models. We did not identify DMRs because we did not know which controls should be needed for the identification of DMRs or DHMRs of these samples.

To address the reviewer's other concerns, we also predicted samples using models based on DMR or DHMR alone. As shown in Supplemental Figure 7, DMR- and DHMR-based models also predicted the brain and liver samples using sscf-MeDIP-seq datasets generated from different amounts of input samples. Furthermore, DMR+DHMR based models performed better than models based on DMRs or DHMRs alone.

Reviewer #1:

5. Processed annotated data should be shared as supplemental information (e.g. values in Fig 2d, 3c, 3f, etc.)

Response: We followed the reviewer's suggestion and put processed annotated data into Source Data using Excel files with corresponding labels.

Reviewer #1

Minor comments:

1. Page 4 "...whole genome DNA methylation is the best.." - the reference suggests that a subset of methylation can perform better than whole-genome DNA methylation. The authors should revise their summary of the reference.

Response: We thank the reviewer for the suggestion and we modified the passage accordingly (p4).

Reviewer #1

2. Page 7 "we analyzed the DNA methylation profiles of liver cancer from TCGA" raw numbers of overlaps between these sets should be reported.

Response: We have reported the raw numbers (Fig. 1d).

Reviewer #1

3. Page 7 "we analyzed the DNA methylation profiles of liver cancer from TCGA" to increase the utility of single-strand methylation analysis the authors should test whether the methylation levels from TCGA were similar to methylation levels in the ss-MeDIP samples (and point out that 450K TCGA samples cannot report strand-specific methylation information).

Response: We would like to point out that our method to analyze DNA methylation is ssg-MeDIP-Seq. In contrast, TCGA utilizes DNA methylation arrays. It is challenging to compare raw values of DNA methylation levels measured using different methods. Therefore, we used an alternative way to compare the similarity between ssg-MeDIP-Seq data and published methylation array data of liver tumor samples (Figure 1d-e).

Reviewer #1

4. Page 8 “we identified 260,055 and 325,866 HMRs in 8 liver tumor samples and their Adj-NT controls, respectively.” Is this the sum of all HMRs or an average per sample? The average per sample should be reported as well. This also applies to reporting of cfDNA DMRs (page 10) and elsewhere where aggregate numbers are reported.

Response: As indicated above, we recalculated the number of HMRs for each sample using the new cutoff. We identified 192,106 and 228,575 HMRs at 8 liver tumors and their corresponding control groups based on the new cutoff. These two numbers represent the number of HMRs at each sample group. To address the reviewer concern, we also used boxplots to show the number of HMRs of each sample in Supplemental Fig. 2d.

Reviewer #1

5. Page 13 – “90% of the training cohort” and “in a balanced way” need to be defined – is this 90% of samples or 90% of features?

Response: We are sorry for the confusion. It is 90% of samples in the training cohort. We modified the text accordingly.

Reviewer #1:

6. Page 14 – Sample selection wording is unclear “Briefly, we randomly chose two cfDNA samples, with each from individuals with liver and brain tumors.”

Response: We chose two cfDNA samples with highest concentration. In this way, we could use different amount of cfDNAs in each sample for analysis. We modified the text accordingly.

Reviewer #1:

7. Page 15 – “Evaluate the sensitivity of the sscf-MeDIP-Seq method.” Were the machine learning outcomes improved if the only-DMR or only-DHMR models were used?

Response: We observed that in general the models trained with DMRs+DHMRs performed more robustly than the models trained with DMRs or DHMRs only (Supplemental Fig. 7).

Reviewer #1:

8. Page 17 – “Together, these results indicate that a significant fraction of liver cancer specific cfDNA DMRs and DHMRs identified in this study are likely associated with changes in the expression of nearby genes in tumor cells, which in turn may contribute to tumorigenesis.” It is unclear what significant fraction this sentence refers to. There is no significance test discussed in this section, and 78/4989 does seem significant enough to make this claim. This sentence should be substantiated, softened or removed.

Response: First, the reviewer is correct that the original sentence is confusing. To clarify the situation in the previous version, we found that the expression of 78 genes in liver cancer tissue samples out of 968 genes with at least one cfDNA DMRs nearby is associated with patient

survival. The number 78 refers to genes, not DMRs. In the revised manuscript, we included 50 additional cfDNA samples in the control group in our analysis. Consequently, the number of genes with at least one liver tumor cfDNA nearby changed. To address this concern, we modified the sentence to avoid the confusion (p17).

Reviewer #1

9. Page 23, DMRs were called in blocks in cfDNA samples, but how were DMRs called in 450K TCGA samples? Were they called as individual probes or somehow expanded to include multiple probes?

Response: The reviewer is correct that the DMRs were called by blocks for scf-MeDIP-Seq datasets. In contrast, DMRs were called at each individual probe for TCGA 450K methylation array datasets. Each individual probe was then tested by Student's t-test and considered as DMR if $p < 0.05$. We did not merge different probes into one.

Reviewer #1

10. Page 25 "Machine learning models" What were the inputs to the models? I assume methylation levels at DMRs and bias at DHMRs, but this should be explicit.

Response: We modified the text accordingly.

Reviewer #1:

11. Page 26 R scripts should be shared via github.

Response: We uploaded the R scripts to GitHub: https://github.com/clouds-drift/plasma_MCD

Reviewer #1:

12. Page 26 "cfDNA DMRs or DHMR are associated with patient survival" hazard ratio cutoff should be provided here.

Response: The hazard ratio cutoff we're using is 1. When the hazard ratio > 1 , the feature is a risky factor; when hazard ratio < 1 , the feature is a protective one.

Reviewer #1:

13. Page 33, Figure 1a – it could be useful to show an example of hemimethylation in the schematic here.

Response: We thank the reviewer for the suggestion and modified Figure 1a accordingly.

Reviewer #1

14. Page 33, Figure 1b, page 7 text "By inspection of MeDIP-seq signals at the gene locus of TBX2, a gene known to be methylated in liver cancer, we identified a DMR specifically in tumors compared to Adj-NT samples" How was this highlighted DMR selected? It looks like there are several other DMRs in this region.

Response: We are sorry for the confusion. We called the DMRs of 8 liver tumor samples based on ssg-MeDIP-Seq datasets and their corresponding Adj-NT samples using the QSEA software (Lienhart et al Nucleic Acid Research (2017), 45, e44). We only showed three samples in Figure 1b to save space. The other regions were not identified as DMRs among these samples based on the QSEA analysis. We modified the text as well as switched the original Figure 1b with Figure 1c to make this point clear.

Reviewer #1:

15. Page 33, Figure 1d – p-value for B is 0 – is this correct?

Response: We added exact p-value to Fig. 1d.

Reviewer #1:

16. Page 33, Figure 1e – Does the LINE value extend beyond the left axis? The x axis should be increased to show the limit of this value.

Response: We modified the previous Figure 1e accordingly (current Figure 1f)

Reviewer #1:

17. Page 33, Figure 1e – N's should be shown for each category as enrichment values are hard to interpret (probably in a supplementary figure if not possible in the main figure)

Response: We added the number of DMRs that show significant enrichment in parenthesis and modified the figure legend.

Reviewer #1:

18. Page 33, Figure 1e – x-axis “Z score enrichment” is not defined. Do the authors mean “Z score”?

Response: It should be Z score and we modified Figure 1e accordingly.

Reviewer #1:

19. Page 35, Figure 2b shows hemimethylation with a scale of -3 to 3. However, the formula implies a range of -1 to 1.

Response: We are sorry for the confusion. Figure 2b represents the RPM value of ssg-MeDIP-Seq datasets. To address this concern, we moved the formula to Figure 2a and indicated the RPM values in Figure 2b legend.

Reviewer #1:

20. Page 35, Figure 2f labels “Increased DHMR” could be changed to “Increased HM” to reflect that the region isn't increasing, but the hemimethylation is changing.

Response: We modified Figure 2f and other figures accordingly.

Reviewer #1

21. Page 42, Figure 1a – how were the IDH subtype model DMRs selected? The heatmap suggests that the DMRs were selected between a different number of samples (~80/20 instead of the 43/34 IDHmut/WT samples described in the text).

Response: DMRs were identified between 43 IDH mutant and 34 IDH WT brain sample groups. The confusion arose from the text in Figure 5a. We modified Figure 5a to avoid confusion (see below).

Reviewer #1

22. Page 42 Figure 5a the schematic of the application of Bayes' theorem is unclear – the boxes make it appear that the $P(\text{IDH mut})$ and $P(\text{IDH WT})$ apply to DMRs while $P(\text{Liver})$ and $P(\text{Control})$ apply to DHMRs.

Response: We are sorry for the confusion. We modified Figure 5a accordingly.

Reviewer #1:

23. Page 44 Figure 6d/g – perhaps x-axis label should be “Time in months” or “Time (months)”?

Response: We made changes in the figures accordingly.

Reviewer #1:

24. Figure S1 – Suggest changing labels to “Promoters with CGI” and “Promoters without CGI.”

Response: We modified the figures accordingly.

Reviewer #2 (Remarks to the Author): expertise in cell free DNA methylation methods

“Sensitive and accurate tumor detection by methylation and hemi-methylation of plasma cell-free DNA” by Hua et al. is a well written article detailing utility of MeDIP-Seq methods. The performance of their classifier models to identify cancer and is promising but it appears that DHMRs do not contribute very much to the performance given especially that there are many more DHMR regions than DMRs.

Response: We thank the reviewer for the precious time to review this manuscript and for insightful comment. In response to the concerns of the three reviewers about hemi-methylation identification, we analyzed 10 input samples including two genomic DNA samples and 8 cfDNA samples without the precipitation by antibodies against 5-mC. In principle, these input samples should not show hemi-methylation. Indeed, these input samples did not show hemi-methylation at the vast majority of ~2M methylation blocks (Supplemental Figure 2). Furthermore, the number blocks showing bias was further reduced using $\text{RPM} > 1$ at each block as the cut off and

was not reduced further from $RMP > 1$ (Supplemental Figure 2). Using $RPM > 1$ as the cut off, we re-analyzed hemi-methylation of all ssg-MeDIP-Seq, and sscf-MeDIP-Seq datasets. We found that the number of DHMRs were significantly reduced. For instance, we identified 24,883 DMRs and 6,562 DHMRs in 8 liver tumor DNA samples compared to their corresponding Adj-NT controls (Figure 2e). These results indicate that many false positive DHMRs were identified previously, likely due to the fact that low sequence reads at each block may impact the DHMR calculation.

In the revised manuscript, we analyzed 50 additional cfDNA samples from the control group. In total, we analyzed cfDNA 271 samples from three groups of samples (control, liver cancer and brain tumor). Using newly identified DHMRs, we also found that models trained with both DMRs and DHMRs showed better predictive performance than models trained with DMRs or DHMRs alone (Figure 4).

We agree with the reviewer that models trained with both DMRs and DHMRs only showed a slight improvement than models using either DMRs or DHMRs alone. The slight improvement is likely due to the fact that DMR-or DHMR-based models alone showed great performance, which makes it challenging to improve further using both DMRs and DHMRs, especially when analyzing 56 samples in the validation cohort. However, we would like to argue that the slight improvement in performance will likely offer benefits in clinical settings when hundreds and thousands of samples would be tested. Second and importantly, one of major issues for early tumor detection is the false positive rate. By evaluating a sample using three different models (DMRs, DHMRs and DMRs+DHMRs), we could predict the results three times independently. If the prediction from three models show concordance, this would increase confidence in the prediction. However, if the predictions from three models show discordance, we could in principle flag the sample for additional tests. Through these approaches, one would expect to reduce false positive prediction and increase prediction accuracy. In the revised manuscript, we discussed these ideas.

Review #2

INTRODUCTION

The statement that targeted bisulfite sequencing “requires up to 80ml of plasma” doesn’t seem accurate. Assuming the authors are referring to Liu et al. which states “Up to 80 ml whole blood was collected from all participants as part of the research study; only two tubes of plasma were processed separately per participant.” 1) that’s 80ml of blood, not plasma and 2) only 2 tubes of blood were analyzed which would yield ~8-10ml of plasma.

Response: We thank the reviewer for pointing out for our oversight and we modified the text accordingly.

Reviewer #2

METHODS/RESULTS

The authors’ ssg-MeDip-Seq ligates different adapters to the Watson and Crick strands, allowing for strand-specific analysis. From the description provided and the diagram (Figure 1A) I don’t see how the same result could not be achieved in silico by doing a standard library

prep and then simply splitting the bam file by strand after alignment. As the authors point out, allele specific methylation is an overlooked and potentially useful biomarker. It is possible to analyze strand specific methylation from bisulfite sequencing; the biscuit BS-Seq aligner, for example, has an allele specific methylation subroutine. With the data that is currently present it is difficult to evaluate the advantage of strand-specific methylation analysis using ss(g/cf)-MeDIP-Seq compared to bisulfite sequencing, as these comparisons are not presented.

Response: We would like to discuss with the following points with the reviewer to address this concern. First, whether a method can detect strand-specific methylation depends on how sequencing libraries are generated. If sequence libraries are prepared using hair-pin adaptors or fork adaptors that mark 3' and 5' end of double-stranded DNA differently (see Letter Figure 2), then the method preserves strand-specific information and consequently could detect hemi-methylation. However, if the library preparation method utilizes the complementary strand (see letter Fig. 2) then strand-specific information will be lost after PCR amplification because double stranded DNA from the original Watson and Crick strand will be the same.

When complementary adaptors are added to the fragments, dsDNA amplified from the Watson and Crick strand are identical, which lose strand-specific information

When fork-headed adaptors are added to a DNA fragment, dsDNAs amplified from the Watson and Crick strand can be distinguished and strand-specificity is preserved

Letter Figure 2. Whether a method can detect strand-specific DNA methylation depends on how the sequence libraries are generated. Top and bottom panels describes a loss and a retention of strand-specific information, respectively.

Second, the reviewer is correct to point out that bisulfite-sequencing (BS-seq) could in principle detect hemi-methylation if the sequencing libraries are prepared using adaptors that preserve strand specific information. In fact, the study that shows DNA hemi-methylation is a novel epigenetic mark utilized a modified BS-seq to analyze DNA methylomes (Xu and Gorce, Science 2018, 359, 1166).

Third, while we did not compare the ssg-MeDIP-seq method with BS-seq directly, we believe that the ssg-MeDIP-seq method has both pros and cons compared to BS-seq. First, the ssg-MeDIP-seq method is relatively simple to perform compared to BS-Seq as the method does not need a library preparation kit and does not need bisulfite conversion. Moreover, BS-seq cannot differentiate between 5-mC and 5hmC. However, unlike the BS-Seq method, ssg-MeDIP-Seq cannot detect methylation at single nucleotide resolution.

Finally, as the reviewer pointed out, DNA hemi-methylation is under studied. In fact, to our knowledge, no studies have used a combination of DNA methylation and hemi-methylation of cfDNA for tumor detection despite the fact that BS-seq could detect hemi-methylation. While we did not understand the reason behind this, one of the authors (ZZ)'s conversation with Dr. Peter

Jones, a pioneer in DNA methylation in cancer, during a seminar visit, offered some insights. Based on his opinions, quoting from his email “hairpin bisulfite sequencing seems to be the best option for detection of hemi-methylation”. However, his lab “has had issues with hairpin bisulfite seq protocols in the past because of low bisulfite conversion rate and other difficulties. Therefore, we are trying to figure out a different way to detect hemi-methylation that does not involve hairpin technology”.

Reviewer #2

The strand specific adaptor ligation shown in figure 1A is absent in figure 3A where they detail the cell-free DNA library prep (sscf-MeDIP-Seq). Is the strand specific adaptor ligation performed in sscf-MeDIP-Seq? Or in this case is strand specific methylation resolved in silico?

Response: Likely due to our writing, we did not make this clear about sscf-MeDIP-seq. Because cfDNA molecules are small in size (100-200bp), and also contain a large fraction of single-stranded DNA (ssDNA) and damaged DNA, we used a different strategy to make sequencing libraries that could preserve strand-specific information for the detection of hemi-methylation. First, we denatured all cfDNA molecules into ssDNAs using heat and then marked the 3' end of each ssDNA molecule with an oligo using a single-stranded DNA ligase. Following synthesis of the second strand, a different adaptor is ligated to the 5'-end. In this way, strand-specific information is preserved. In the revised manuscript, we modified Figure 3A to make this clear. In short, hemi-methylation detected by sscf-MeDIP-Seq is not resolved in silico.

Reviewer #2.

The authors state that their sscf-MeDIP-Seq library prep is superior to other cfDNA methods in part because it can recover single stranded and damaged DNA, which is particularly important for cfDNA applications. However, the authors do not compare their method with traditional MeDIP which begs the question: how much is gained by performing sscf-MeDIP-Seq versus previously published methods and studies on cfDNA MeDIP-Seq (PMID: 35065650, 31471598, 31471598)? Does it improve sequencing output given identical input? Does it significantly improve signal detection? A direct comparison to previously published methods would strengthen the results.

Response: We agree with the reviewer that it would be ideal to perform a direct comparison of sscf-MeDIP-Seq datasets with published cfDNA MeDIP-seq. However, we could not access these datasets even after we sent multiple requests to access these published datasets in the last three and half years. The most recent request was sent on November 16, 2023. While we could in principle follow the published cfDNA MeDIP-seq to generate datasets for comparison, we did not do this for the following reasons. First, it took us a long time to optimize sscf-MeDIP-Seq procedures and we suspect that it may take a long time to follow published procedures to generate reliable MeDIP-Seq datasets. Second and importantly, we have analyzed over 300 samples using sscf-MeDIP-seq. Therefore, it may not be a good comparison if we are very good at handling sscf-MeDIP-seq procedures than cf-MeDIP-seq procedures. Importantly, our method can detect hemi-methylation. We are not sure the published cfDNA MeDIP-Seq can. Even if the method could, the published study did not use hemi-methylation for tumor detection.

To address this concern further, we searched the literature and found a couple of published studies comparing different cfDNA library procedures. They found that the single-stranded cfDNA preparation method, which we used here, is more sensitive than the traditional double stranded DNA preparation method (Burnham et al Scientific Reports 6: *Sci Rep* 6, 27859 (2016)). In the revised manuscript, we cited these papers and modified the discussion extensively to avoid direct comparison between sscf-MeDIP-Seq and published cf-MeDIP-Seq.

Reviewer #2.

Sequencing metrics that might be used to evaluate ss(g/cf)-MeDIP-Seq are absent (total reads, alignment rate, duplication rate etc.). It would also be useful to know the sequencing depth at each of the ~2M CpG clusters the authors evaluated.

Response: In response to the reviewer’s request, we have compiled “**Table 1 Sample and Reads info**” listing total reads, alignment rate and duplication rate of all samples used in this study and put them the Source Data

We also calculated the average sequencing depth of 2M CpG clusters of 10 cfDNA case samples and 10 control samples (Letter Fig. 3).

Letter Figure 3. Mean raw read count and Mean RPM at ~2M blocks of 10 liver cancer cfDNA samples and 10 controls used in Figure 3.

Reviewer #2

*The overlap between DMRs and DHMRs indicates that they are largely independent. However hemi-methylation would be ~50% methylated, if only considering **beta value**, and therefore hypomethylated compared to average genomic methylation levels; I would expect more overlap. If you reduce the stringency of the filtering criteria for DMR analysis, do you see an increase in the overlap?*

Response: We thank the reviewer for the insightful comments. As detailed above, we reanalyzed hemi-methylation of all samples by including an additional cut off (RPM > 1) based on the assumption that input samples without the methylated DNA immunoprecipitation step showed little, if any, strand-specific bias. Using additional cutoff, we found that the number of DHMRs was reduced markedly. However, while the overlap between DHMRs and DMRs increased (Figure 2e), 4,474 of 6,562 liver tumor DNA DHMRs did not overlap with DMRs. Similar

analysis on cfDNA DHMRs from 10 liver tumor samples compared to 10 controls also reveals that a large fraction of DHMRs did not overlap with DMRs (Figure 3g). These results are consistent with published studies that DNA hemi-methylation is an independent epigenetic mark (Xu and Gorce, Science 2018, 359, 1166; and Thomas et al, Nucleic Acid Research 2023, 51: 5997-6005).

To address this concern further, we calculated DNA methylation density of Watson and Crick strands at liver tumor specific DMRs and DHMRs compared to controls. We found that DNA methylation density of both Watson and Crick strand at DMRs were changed almost equally, whereas DNA methylation density at DHMRs were changed markedly only at one strand (Supplemental Figure 3), providing an explanation for the observations that most DHMRs did not overlap with DMRs.

Reviewer #2

In the model training, the authors “selected the top 100 DMRs and 741 DHMRs” for model training/testing. How were cutoffs determined (n = 100 DMR & n = 741 DHMR)? Seems a bit lopsided in favor of the DHMRs, which do not seem to be adding much to the performance of the model. The authors state that the DMR+DHMR model performs better than either DMR or DHMR models alone but the DMR model has 100 features, no? What’s the performance of the DMR model if you take the top 841 DMRs, so the total number of features is equal? Or if you were to train using the top X DMRs and top X DHMRs – this would allow for more direct comparison between the feature types.

Response: As detailed above, based on analysis of input samples, we included an additional cutoff (RPM>1 at each block) to analyze hemi-methylation. We found far fewer DHMRs for each group of samples. In the revised manuscript, we also followed the reviewer’s suggestion and used an equal number of DMRs and DHMRs (200 each) for as inputs to machine learning. These DMRs and DHMRs were selected based on the feature importance of 215 samples in the training cohort. Please note that we analyzed 50 more cfDNA samples from the control group in the revised manuscript. Under these conditions, we found that 1) models trained with DHMRs alone performed better than DHMR models of last submission, and 2) DMR+DHMR models performed a slight better than models using DMRs or DHMRs alone (Figure 4). As discussed above, the slight improvement is likely due to high performance DMR- or DHMR models, which makes it challenging to improve the performance when DMRs+DHMRs are combined to analyze the 56 samples in the validation cohort.

We also selected the top 100 DMRs and DHMRs to train machine learning models. The models could also predict samples in the validation cohort very well (Letter Fig. 4), revealing the robustness of prediction.

Letter Figure 4. Predicting tumor types by models trained with DMRs, DHMRs and DMRs+DHMRs.

Evaluation of model performances for the prediction of control (A), liver tumor (B) and brain tumor (C) cfDNA samples in the validation cohort using models trained with 100 DMRs, 100 DHMR, or 100 DMRs+ 100 DHMRs. The best sensitivity and specificity point for each prediction were marked with the Red dot. (D) The average prediction

probability of each group of samples using models trained with DMRs+DHMRs. Each column represents the group of validation samples, with each row representing model predictions. Red, yellow and blue bars represent probability of samples being from brain cancer, liver cancer, and healthy controls, respectively.

Reviewer #2

Did you force the cluster breaks in the heatmaps in figure 6? C/F appear to have been forcefully split by row and column groups using the `row_split` and `column_split` options, assuming the authors are used the `ComplexHeatmap` R package. If so, this should be stated.

Response: Figure 6 C/F is clustered by unsupervised clustering using the `pheatmap` package (<https://cran.r-project.org/web/packages/pheatmap/index.html>). We did not forcefully split by row and column groups, nor did we use the `ComplexHeatmap` package.

Review #2

ROC curve AUCs should have confidence intervals either stated in the text or annotated on the figure, ideally both.

Response: We added the confidence intervals in the text, figure and figure legends.

Reviewer #2

DISCUSSION

The authors state many advantages of their methods, please provide direct comparisons to previously published methods.

Response: As discussed above, while we would be happy to compare our methods with other published methods, we could not. In the revised manuscript, we modified the text to avoid direct comparison of sscf-MeDIP-Seq with other published methods.

*Reviewer #3 (Remarks to the Author): expert in machine learning cfDNA analysis
The author employed an enhanced MeDIP-Seq technique to examine DNA methylation patterns in liver cancer and brain cancer, underscoring the effectiveness of utilizing both DMRs and DHMRs for accurate cancer detection. Nevertheless, in previous studies, bioinformatics approaches for stranded methylation detection and hemi-methylation region identification from MeDIP-Seq data have already been well-established. Furthermore, both MeDIP-Seq sequencing technology and the utilization of the Tn5 enzyme for fragmenting and tagging double-stranded DNA in Next-Generation Sequencing (NGS) are well-established, mature techniques, emphasizing a notable lack of significant innovation.*

Response: We thank the reviewer for spending precious time to review the manuscript. The reviewer raised several points. I would like to discuss the following points with the reviewer. First, I agree with the reviewer that previous studies on DNA methylation in cell lines and/or in other animal species, using either MeDIP-seq and BS-seq datasets, could detect DNA hemi-methylation. However, no studies, to our knowledge, analyzed both DNA methylation and hemi-methylation of plasma cell free DNA for tumor detection, which is the focus of the present study. Furthermore, our studies show for the first time that most DHMRs do not overlap with DMRs based on analysis of tumor DNA and cfDNA samples from liver cancer patients. Finally, we would like to point out that our study, to our knowledge, is the first to combine the existing technologies to study DNA methylation in a strand-specific way. For instance, while Tn5 has been used to fragment DNA for NGS library preparation, few studies, if any, have combined DNA fragmentation by Tn5 with methylated DNA immunoprecipitation to analyze DNA methylation and DNA hemi-methylation. In short, our studies, like most studies in the literature, benefit from these published studies.

Reviewer #3

Hemi-methylated DNA typically has a propensity to either become fully methylated or tend towards demethylation. Varying DMR identification thresholds can be used to extract methylation change data in these regions. Moreover, the gold standard for methylation identification, WGBS (Whole Genome Bisulfite Sequencing), can differentiate between the positive and negative strands to acquire strand-specific methylation changes. This highlights a notable lack of impact.

Response: First, as stated above, BS-seq could in principle detect hemi-methylation if the library preparation method preserves strand-specific information. Second and importantly, while hemi-methylation could be detected by BS-seq, our study is the first to utilize both cfDNA DMRs and DHMRs for tumor detection. Further, we show for the first time that DMRs and DHMRs are likely independent biomarkers. Third, as pointed out above, we could predict one sample using three different machine learning models (DMRs, DHMRs, DMRs+DHMRs). If all three models predict the same outcome, this will increase confidence in tumor detection. However, if the three

models show discordance, this in principle allows us to be cautious about the prediction. In principle, this will likely reduce the false positive predictions, a major challenge in liquid biopsy. We discussed these points in the discussion.

Reviewer #3

My main detailed concerns regarding this manuscript are as follows:

1. Hemi-methylated regions represent a relatively small fraction of the genome, yet the number of DHMRs (Differentially Hemi-Methylated Regions) is significantly greater than the count of DMRs (Differentially Methylated Regions). This raises questions about whether the threshold set for DMR identification may have resulted in some regions that could potentially be identified as DMRs going unnoticed, ultimately leading to a limited overlap between DHMRs and DMRs. This undermines the conclusion that the factors contributing to DHMRs are independent of those associated with DMRs.

Response: We thank the reviewer for the insightful comments. As described in the responses to the concerns of both reviewer #1 and reviewer #2, we found that DHMRs identified previously most likely contained a large number of false positive ones. In short, we sequenced 10 input samples without being subjected to methylated DNA immunoprecipitation and analyzed “hemi-methylation” or bias at ~2 M methylation blocks of these input samples. In principle, these input samples should not show any hemi-methylation. Indeed, we found that these input samples did not show bias at the vast majority of blocks. Using $RPM > 1$ at each block as the cut off, we found that the number of blocks showing bias are reduced further, suggesting that sequencing depth may affect the identification of hemi-methylated regions. Using $RPM > 1$, we re-analyzed hemi-methylation of all ssg-MeDIP-Seq and sscf-MeDIP-seq datasets. We found that the number of hemi-methylation sites decreased markedly. Consequently, we found that DMRs far exceed DHMRs in both tumor DNA samples (Figure 2e) and cfDNA samples (Figure 3g).

Reviewer #3

2. MeDIP-seq exhibits a proclivity for interrogating genomic regions characterized by low CpG density, and fewer CpG sites are more susceptible to sequencing technology errors and random inaccuracies, leading to bias in identifying hemi-methylated regions.

Response: To address this concern, we used the same procedures and prepared the libraries of input samples, which were not subjected to methylated DNA immunoprecipitation, for sequencing. If bias arises from sequencing technology errors and random inaccuracies, one would expect that the same bias could also be also detected at these input samples. We found that these input samples did not show bias at the vast majority of methylation blocks. Furthermore, with $RPM > 1$ at each block as the cutoff, we identified fewer biased regions compared to $RPM > 0.5$, and a further increase in the number of reads at each block did not reduce the number of blocks showing bias in these input samples markedly. Therefore, using $RPM > 1$ as an additional cutoff, we could markedly minimize potential false positive hemi-methylated regions. In the revised manuscript, we reanalyzed HMRs of sscf-MeDIP-seq datasets and found that machine learning models trained with DHMRs performed better than models trained with DHMR identified previously. For instance, the AUCs for control, liver and brain tumor samples were 0.761, 0.933 and 0.885, respectively, based on models trained with DHMRs identified before (Figure 4 of previous version). The AUCs for control, liver and brain tumor

samples were 0.899, 0.954 and 0.908, respectively, based on DHMRs identified in the revised manuscript (Figure 4), supporting the idea that DHMRs identified using the new cutoff are likely more accurate than those identified last time.

Reviewer #3

3. The accuracy and sensitivity of cfDNA abnormal methylation in cancer detection have been previously reported in earlier studies. And the title is too generic and fails to highlight the main content and innovative aspects of the article.

Response: We modified the title to highlight the novelty of this study: Tumor detection by analysis of both symmetric- and hemi-methylation of plasma cell free DNA.

Reviewer #3:

4. External cohorts are needed to validate the diagnostic accuracy of DHMRs for lung and liver cancer, thereby preventing overfitting in model development.

Response: We agree with the reviewer that it would be great to test our models for liver and brain tumor using an external cohort of samples. We would like to point out that it is very challenging to obtain these kinds of samples externally. For instance, we established a collaboration with Dr. Scott Kaufmann at Mayo Clinic on ovarian cancer. These samples have been collected already at Mayo Clinic many years ago. It took 6 months for Columbia University and Mayo Clinic to come up with an agreement on how to share potential intellectual properties. Therefore, it is very challenging to obtain an independent cohort of cfDNA samples from brain and liver cancer patients from an independent source.

However, through the collaboration efforts on ovarian cancer project, we obtained 34 control samples. We found that our machine learning models including DHMR models predicted these samples extremely accurately (Letter Figure 5). While these samples validated our models for the controls, we decided not to put these results in this manuscript for two reasons. First, we needed to ask permission of Dr. Kaufmann to include these samples. Second and importantly, these samples are not sex balanced as they are all from women. We are not sure whether this will affect the prediction outcome. Nonetheless, these results at least show that our models trained with DHMRs most likely provide additional diagnostic accuracy.

Letter Figure 5: Predict the outcome of 34 control samples from an independent cohort using DMR-, DHMR-, and DMR+DHMR-based models. (A) Evaluation of model performances for the prediction of being a control. The best sensitivity and specificity point for each

prediction were marked with a red dot. (B) The average prediction probability using models trained with DMRs, DHMRs and DMRs+DHMRs (Final). These 34 samples were collected by the Mayo Clinic as the control samples for an independent study on ovarian cancer. The models were trained using 215 cfDNA samples described in Figure 4.

Reviewer #3:

5. In the clinical setting, blood tests for late-stage cancer patients are not meaningful. What is the distribution of cancer stages in your cohort? It is necessary to separately examine the diagnostic accuracy of early-stage DHMRs to demonstrate their practical significance in cancer diagnosis.

Response: We separated liver cancer samples in the validation cohort into early and late groups and found that our model predicted early and late stages of liver cancer equally well (Supplemental Fig. 6).

Reviewer #3:

6. Raw data and code should be provided to ensure that data availability allows independent verification of results and increases the transparency of scientific research.

Response: All sequencing datasets were deposited at dbGAP (see link below). This study was not supported by NCI, and it took us a long time to go through the approval process at Columbia University and NCI so that we could deposit data related to human samples to dbGaP. (https://www.ncbi.nlm.nih.gov/projects/gap/cgi-bin/study.cgi?study_id=phs003462). The custom code was uploaded to github: https://github.com/clouds-drift/plasma_MCD.

REVIEWER COMMENTS

Reviewer #1 (Remarks to the Author):

The authors' responses appear to sufficiently address the reviewers' concerns. I am pleased to see that a more stringent cutoff of coverage reduced false-positives and improved machine learning outcomes.

Reviewer #1 (Remarks on code availability):

Code provides README file with instructions for running code. Code is decently annotated. The authors could improve the repository by including column names in provided data files.

Reviewer #2 (Remarks to the Author):

I would like to thank the authors for performing a thorough and detailed revision of their manuscript; the revised version and rebuttal letter addresses many of the questions and comments I had with the original submission and is a much stronger paper overall. However, there remains several other important issues that need to be resolved.

RESULTS

In the section where the authors evaluate the sensitivity of sscf-MeDIP-Seq they use different input for the brain cancer sample (3.5,10,24ng) and liver cancer sample (3,7,15ng) (Suppl Fig. 7). Why exactly are the input amounts inconsistent between the two samples?

The authors state they could obtain data from 'as little as 20 ul plasma.' I am assuming you are referring to the lower limit of their input experiment (3ng cfDNA). As I am sure the authors are aware, the concentration of plasma cfDNA can vary quite a lot between individuals – I think this statement should be further qualified.

METHODS

I would like to thank the authors for providing a GitHub repository with code and data for independent researchers to evaluate the results. I have a few comments regarding the repo:

I assume the `*score.txt` files appear to contain the raw data used to construct the machine learning models provided on the github repository. Merging the data in the DHMR score files shows a large proportion of missing (NA) values (~40%). Comparing these data to the training data in the ML models on the repository, it appears all NA values have been imputed as 0 in the ML models. I assume the DHMR models use the 'bias' calculation as ML input since all features are numeric values between -1 and 1, which matches the description of 'bias' ((Watson-Crick)/(Watson+Crick)). By this logic, the NA values would potentially be due to $\text{Watson+Crick} = 0$, which would mean no reads were mapped to the region. If I understand correctly, this could be due to a lack of methylated cytosines at this locus (i.e., completely hypomethylated) leading to no IP pulldown, and therefore there would not be a bias, in which case it would be appropriate to assign these regions a score of 0. An alternate explanation, however, is that no reads are mapped because of inadequate sequencing, or an inefficient IP.

A few points: 1) the authors should describe the data processing and machine learning input in more detail, which would remove a lot of ambiguity here. 2) Have the authors considered technical variables that could influence the number of reads mapped, and potentially the frequency of NA values, to DHMR regions? 3) Do the authors believe the lower performance of DHMR models could be due to this imputation?

There appear to be 41 files in both the DMR and DHMR directories for samples that were not included in the training or validation sets (based of the excel sheets). Were these samples used in some other part of the manuscript for a different purpose? They appear to mostly be normal, "ZH680_normal_M19035001586" is one example.

DISCUSSION

I had said, in the first round of peer review, "The authors state many advantages of their methods, please provide direct comparisons to previously published methods" and the authors stated in the rebuttal that they 'were unable to directly compare their method to other published methods.' I should have been clearer in the original comment – I was talking about the overall performance of their method/models to other studies also performing cancer classification using cfDNA. The cfDNA results are very good, putting those results in the context of other studies would strengthen the discussion.

The authors state "by evaluating samples using three different models (DMRs, DHMRs and DMRs+DHMRs), we could envision to reduce false positives during cancer screening, a major issue for early tumor detection using current assays. Because we could predict the same sample three times independently, we could in principle flag the sample for additional tests if predictions from three models show discordance." Do the authors have any examples in the validation set where a false positive would be reclassified as a true negative using this approach?

Reviewer #2 (Remarks on code availability):

included with main comments

Reviewer #3 (Remarks to the Author):

Upon reviewing the revised manuscript, it's evident that the authors have effectively addressed the specific issues highlighted in the initial feedback. The modifications made, particularly the enhancement of the title to better reflect the study's innovative aspects, the incorporation of an external validation cohort, and the adjustment of thresholds to address the issue of excessive hypermethylated regions, clearly indicate a focused effort to refine the study based on the provided recommendations.

The decision to include an external validation cohort enriches the study by offering additional evidence of the model's robustness, effectively mitigating concerns regarding overfitting. This addition strengthens the manuscript's claims and broadens its appeal.

The adjustment of thresholds to manage the identification of hypermethylated regions demonstrates a proactive approach to addressing theoretical concerns, ensuring that the analysis remains aligned with established scientific principles. This adjustment likely enhances the accuracy and relevance of the findings, making the study more valuable to readers and researchers interested in the field.

Given the revisions undertaken, the manuscript now presents a clearer, more compelling narrative that is likely to engage a wider audience. The steps taken to address the initial feedback not only improve the manuscript's clarity and precision but also highlight the authors' dedication to presenting their research in the best possible light.

In conclusion, the revised manuscript represents a significant improvement over the original submission. The authors have taken careful steps to address the concerns raised, resulting in a study that is well-positioned to make a meaningful contribution to its field.

REVIEWER COMMENTS

Reviewer #1 (Remarks to the Author):

The authors' responses appear to sufficiently address the reviewers' concerns. I am pleased to see that a more stringent cutoff of coverage reduced false-positives and improved machine learning outcomes.

Response: We thank the reviewer for the time reviewing the manuscript and for the support of publication of this important study.

Reviewer #1 (Remarks on code availability):

Code provides README file with instructions for running code. Code is decently annotated. The authors could improve the repository by including column names in provided data files.

Response: Thanks for the reviewer's suggestion. We added the column names for the following files.

“QSEA_diff/215_training_set_other_nature_block_QSEA/all_sample/pvalue0.01_LFC1/total200.bed” and

“DMBR_diff/215_training_set_other_nature_block_RPM1_each0.3_miss1/all_sample/pvalue0.01_delta0.3_base0.3/total200.bed”.

Reviewer #2 (Remarks to the Author):

I would like to thank the authors for performing a thorough and detailed revision of their manuscript; the revised version and rebuttal letter addresses many of the questions and comments I had with the original submission and is a much stronger paper overall. However, there remains several other important issues that need to be resolved.

Response: We thank the reviewer for the time reviewing the manuscript and for the very positive comments. We have attempted to address each of the concerns detailed below.

Reviewer #2

RESULTS

In the section where the authors evaluate the sensitivity of sscf-MeDIP-Seq they use different input for the brain cancer sample (3.5,10,24ng) and liver cancer sample (3,7,15ng) (Suppl Fig. 7). Why exactly are the input amounts inconsistent between the two samples?

The authors state they could obtain data from ‘as little as 20 ul plasma.’ I am assuming you are referring to the lower limit of their input experiment (3ng cfDNA). As I am sure the authors are aware, the concentration of plasma cfDNA can vary quite a lot between individuals – I think this statement should be further qualified.

Response: Thank the reviewer for pointing this out. As the reviewer stated above, the amount of cfDNA purified from each plasma sample can vary a lot among individual samples. Therefore, we did not use exact amount of cfDNA for sscf-MeDIP-Seq. Instead, we normally used 1/3 to 1/2 cfDNA purified from 1 ml plasma samples for our analysis. Therefore, when we tested different amount of cfDNA for the generation of sscf-MeDIP-Seq datasets, we used different fractions of

cfDNA purified from each sample instead of the exact amount of cfDNA. In the revised manuscript, we made this clear (p16).

To answer the reviewer's inquiry precise, the three amounts of cfDNA (3.5 ng, 10 ng, 24 ng) for the brain tumor sample are equivalent to 21 μ l, 63 μ l and 150 μ l of plasma used for purification. The three different amounts of cfDNA of the liver cancer sample (3 ng, 7 ng and 15 ng) are equivalent to 60 μ l, 140 μ l, and 300 μ l plasma of the sample. As described in the manuscript, these two samples were chosen for analysis because they contained high concentration of cfDNA, which allowed us to perform sscf-MeDIP-seq using different amount materials. Therefore, these two samples do not represent the majority of cfDNA samples. To avoid potential confusion, we deleted the sentence from the revised manuscript (p16).

Reviewer #2

METHODS

*I would like to thank the authors for providing a GitHub repository with code and data for independent researchers to evaluate the results. I have a few comments regarding the repo: I assume the “*score.txt” files appear to contain the raw data used to construct the machine learning models provided on the github repository. Merging the data in the DHMR score files shows a large proportion of missing (NA) values (~40%). Comparing these data to the training data in the ML models on the repository, it appears all NA values have been imputed as 0 in the ML models. I assume the DHMR models use the ‘bias’ calculation as ML input since all features are numeric values between -1 and 1, which matches the description of ‘bias’ ((Watson-Crick)/(Watson+Crick)). By this logic, the NA values would potentially be due to Watson+Crick = 0, which would mean no reads were mapped to the region. If I understand correctly, this could be due to a lack of methylated cytosines at this locus (i.e., completely hypomethylated) leading to no IP pulldown, and therefore there would not be a bias, in which case it would be appropriate to assign these regions a score of 0. An alternate explanation, however, is that no reads are mapped because of inadequate sequencing, or an inefficient IP.*

Response: The reviewer is correct that the file labeled score.txt contains the bias score of sscf-MeDIP-Seq. In this revised manuscript, we annotated the Score.txt in the GitHub repository for “DMR_model/score” and “DHMR_model/score” files to make it clearer.

The author's explanation for potential reasons for the blocks labeled with the NA is also correct. As mentioned in the method section, we used RPM>1 at each methylation block as the cutoff based on analysis of input samples. Therefore, if a block does not meet this criterion, the block in the sample will be labeled as “NA” in the score sheet. In the revised manuscript, we followed the reviewer suggestions and added a couple of sentences describe the NA in the method section (p28, detailed below).

At the same time, I would also like to point out that DHMR are regions showing significantly differential sscf-MeDIP-Seq bias score among different groups of samples. The block labeled with “NA” in one sample may not be “NA” in other samples of the same group. Therefore, the inadequate sequencing depth in one sample, if not occurring in other samples of this group, will unlikely affect DHMR identifications for this sample group.

Reviewer #2

A few points: 1) the authors should describe the data processing and machine learning input in more detail, which would remove a lot of ambiguity here. 2) Have the authors considered technical variables that could influence the number of reads mapped, and potentially the frequency of NA values, to DHMR regions? 3) Do the authors believe the lower performance of DHMR models could be due to this imputation?

Response: We followed the reviewer's suggestion and describe the data processing and machine learning input in more detail in the revised manuscript (p28). Second, we shared the reviewer's concern that technical variables may influence the numbers of reads mapped, and potentially the frequency of NA values for each sample. Based on our experience through analysis of close to 300 cfDNA sscf-MeDIP-Seq datasets, as the reviewer may predict, the main variable is the amount of cfDNAs used for sscf-MeDIP-Seq experiment. However, this issue, I suspect, that may also affect the identification of both DMRs and DHMRs. Importantly, I believe that it is a general problem that we need to deal with no matter which methods are used to analyze plasma cfDNA methylomes for tumor detection.

Third, I also agree with the reviewer that the imputation for identification of DHMRs likely contributes to the lower performance of DHMR models compared to DMRs models. In the revised manuscript, we added this idea to the method section after describing the caveats for our analysis of DHMRs (p28).

Reviewer #2

There appear to be 41 files in both the DMR and DHMR directories for samples that were not included in the training or validation sets (based of the excel sheets). Were these samples used in some other part of the manuscript for a different purpose? They appear to mostly be normal, "ZH680_normal_M19035001586" is one example.

Response: As shown in Letter Fig. 5 in previous response letter, we used the models trained in this study to predict 34 controls samples collected from Mayo Clinic. These 34 samples were not included in the manuscript, but the scores for these 34 samples were put in the repository for reviewers to review them. Four files were from different amounts of cfDNA (Figure S7). We removed three other files that were not used in this manuscript and were put in the Table by oversight. If the reviewer likes, we could remove all these 41 files.

Reviewer #2

DISCUSSION

I had said, in the first round of peer review, "The authors state many advantages of their methods, please provide direct comparisons to previously published methods" and the authors stated in the rebuttal that they 'were unable to directly compare their method to other published methods.' I should have been clearer in the original comment – I was talking about the overall performance of their method/models to other studies also performing cancer classification using cfDNA. The cfDNA results are very good, putting those results in the context of other studies would strengthen the discussion.

Response: Thank the reviewer for the clarification of previous concern. We followed the reviewer's suggestion and compared our methods with two highly-cited studies by analysis of methylomes of cfDNA using different methods (p21-p22).

Reviewer #2

The authors state "by evaluating samples using three different models (DMRs, DHMRs and DMRs+DHMRs), we could envision to reduce false positives during cancer screening, a major issue for early tumor detection using current assays. Because we could predict the same sample three times independently, we could in principle flag the sample for additional tests if predictions from three models show discordance." Do the authors have any examples in the validation set where a false positive would be reclassified as a true negative using this approach?

Response: The short answer is yes. For instance, of the 21 control (normal) samples in the validation cohort, 3 samples were predicted by the DMR models as "brain tumor" and as "Normal" by DHMR-based models. In theory, we could flag these samples for additional analysis.

Reviewer #3 (Remarks to the Author):

Upon reviewing the revised manuscript, it's evident that the authors have effectively addressed the specific issues highlighted in the initial feedback. The modifications made, particularly the enhancement of the title to better reflect the study's innovative aspects, the incorporation of an external validation cohort, and the adjustment of thresholds to address the issue of excessive hypermethylated regions, clearly indicate a focused effort to refine the study based on the provided recommendations. The decision to include an external validation cohort enriches the study by offering additional evidence of the model's robustness, effectively mitigating concerns regarding overfitting. This addition strengthens the manuscript's claims and broadens its appeal.

The adjustment of thresholds to manage the identification of hypermethylated regions demonstrates a proactive approach to addressing theoretical concerns, ensuring that the analysis remains aligned with established scientific principles. This adjustment likely enhances the accuracy and relevance of the findings, making the study more valuable to readers and researchers interested in the field.

Given the revisions undertaken, the manuscript now presents a clearer, more compelling narrative that is likely to engage a wider audience. The steps taken to address the initial feedback not only improve the manuscript's clarity and precision but also highlight the authors' dedication to presenting their research in the best possible light.

In conclusion, the revised manuscript represents a significant improvement over the original submission. The authors have taken careful steps to address the concerns raised, resulting in a study that is well-positioned to make a meaningful contribution to its field.

Response: We thank the reviewer for the time reviewing the manuscript and for the support of publication of this study.

REVIEWERS' COMMENTS

Reviewer #2 (Remarks to the Author):

We thank the authors for the edits. We have no further questions.